# LLM-SR: Scientific Equation Discovery via Programming with Large Language Models

**Parshin Shojaee**[1][*]    **Kazem Meidani**[2][*]    **Shashank Gupta**[3]
**Amir Barati Farimani**[2]    **Chandan K. Reddy**[1]

[1]Virginia Tech    [2]Carnegie Mellon University    [3]Allen Institute for AI

## Abstract

Mathematical equations have been unreasonably effective in describing complex natural phenomena across various scientific disciplines. However, discovering such insightful equations from data presents significant challenges due to the necessity of navigating extremely large combinatorial hypothesis spaces. Current methods of equation discovery, commonly known as symbolic regression techniques, largely focus on extracting equations from data alone, often neglecting the domain-specific prior knowledge that scientists typically depend on. They also employ limited representations such as expression trees, constraining the search space and expressiveness of equations. To bridge this gap, we introduce LLM-SR, a novel approach that leverages the extensive scientific knowledge and robust code generation capabilities of Large Language Models (LLMs) to discover scientific equations from data. Specifically, LLM-SR treats equations as programs with mathematical operators and combines LLMs' scientific priors with evolutionary search over equation programs. The LLM iteratively proposes new equation skeleton hypotheses, drawing from its domain knowledge, which are then optimized against data to estimate parameters. We evaluate LLM-SR on four benchmark problems across diverse scientific domains (e.g., physics, biology), which we carefully designed to simulate the discovery process and prevent LLM recitation. Our results demonstrate that LLM-SR discovers physically accurate equations that significantly outperform state-of-the-art symbolic regression baselines, particularly in out-of-domain test settings. We also show that LLM-SR's incorporation of scientific priors enables more efficient equation space exploration than the baselines[1].

## 1 Introduction

The emergence of Large Language Models (LLMs) has marked a significant milestone in artificial intelligence, showcasing remarkable capabilities across various domains (Achiam et al., 2023). As LLMs continue to evolve, researchers are exploring innovative ways to harness their potential for solving complex problems such as scientific discovery (Wang et al., 2023a; AI4Science & Quantum, 2023). Their ability to process and comprehend vast amounts of scientific literature, extract relevant information, and generate coherent hypotheses has recently opened up new avenues for accelerating scientific progress (Zheng et al., 2023b; Ji et al., 2024). Additionally, by leveraging their ability to understand and reason with the help of programming and execution, LLMs have shown the potential to enhance automated reasoning and problem-solving capabilities for general natural language and mathematics optimization tasks, e.g., prompt optimization and heuristic discovery (Meyerson et al., 2023; Yang et al., 2023a; Madaan et al., 2024; Romera-Paredes et al., 2024). Motivated by these strengths, LLMs could be particularly helpful for the task of equation discovery, a fundamental task in science and scientific discovery.

Discovering accurate symbolic mathematical models from data is an important task in various scientific and engineering disciplines. The task of *data-driven equation discovery* (also commonly known as Symbolic Regression (SR)), aims to find abstract mathematical equations from data

---

[*]Equal contribution. Contact: parshinshojaee@vt.edu, mmeidani@andrew.cmu.edu
[1]Code and data are available: `https://github.com/deep-symbolic-mathematics/LLM-SR`

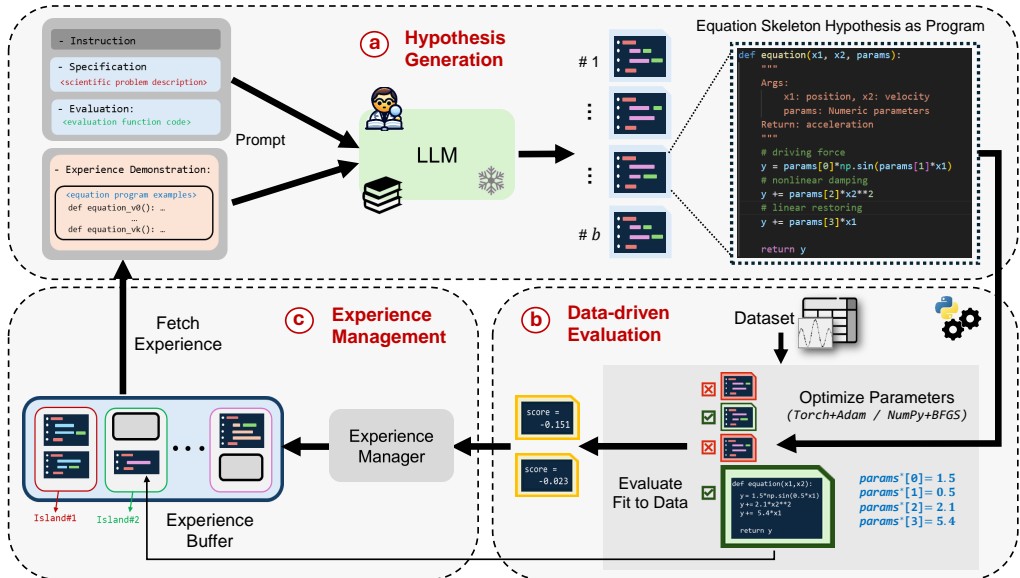

Figure 1: **The LLM-SR framework**, consisting of three main steps: **(a) Hypothesis Generation**, where LLM generates equation program skeletons based on a structured prompt; **(b) Data-driven Evaluation**, which optimizes the parameters of each equation skeleton hypothesis and assesses its fit to the data; and **(c) Experience Management**, which maintains a diverse buffer of high-scoring hypotheses to provide informative in-context examples into LLM's prompt for effective iterative refinement.

observations such that these equations are predictive of the underlying data, are interpretable, and generalize to unseen data from the same physical phenomena. Finding such equations offers several advantages over simply estimating a predictive model, as the resulting mathematical functions provide insights into the underlying physical processes, enable extrapolation beyond the observed data, and facilitate knowledge transfer across related problems (Langley, 1981; Schmidt & Lipson, 2009). However, while evaluating the fit of a proposed equation is relatively straightforward, the inverse process of obtaining these mathematical equations from data is a challenging problem, known to be NP-hard (Virgolin & Pissis, 2022). Current equation discovery methods encompass a wide variety of approaches from evolutionary search algorithms (Cranmer, 2023; Mundhenk et al., 2021; La Cava et al., 2021) to advanced deep learning methods using Transformers (Biggio et al., 2021; Kamienny et al., 2022). Most of the traditional symbolic regression techniques are built on top of Genetic Programming (GP) (Koza, 1994) evolutionary methods, representing mathematical equations as expression trees and searching the combinatorial space of possible equations through iterative mutation and recombination. However, these methods often struggle with the complexity of the vast optimization space and do not incorporate prior scientific knowledge, which leads to suboptimal solutions and inefficient exploration of the equation search space. Similarly, the design of current general LLM-based optimization frameworks (Meyerson et al., 2023; Romera-Paredes et al., 2024; Yang et al., 2023a) also have several key limitations in terms of domain knowledge integration and diverse exploration which are critical for equation discovery. Thus, *there is a need for specialized equation discovery methods that effectively integrate prior scientific knowledge into the navigation of vast equation search space, a strategy akin to a scientist's reliance on foundational scientific knowledge when formulating hypotheses for scientific discovery.*

To address all these limitations, we introduce LLM-SR (shown in Fig. 1), a novel framework that combines the strengths of LLMs, reliable optimizers, and evolutionary search for data-driven equation discovery. At its core, LLM-SR is an iterative hypotheses refinement method that generates, evaluates, and refines equation hypotheses based on data-driven feedback. Specifically, LLM-SR first prompts the LLM to propose new equation hypotheses (Fig. 1(a)), then evaluates their fit on the observed data using off-the-shelf optimizers (Fig. 1(b)), and uses this data-driven feedback and a carefully maintained dynamic memory of previous equations (Fig. 1(c)) to iteratively guide the search towards better equations. LLM-SR leverages the scientific knowledge embedded in LLMs using short descriptions of the problem and the variables involved in a given system to generate educated

hypotheses for equation skeletons (i.e., mathematical structures with placeholder parameters for numeric coefficients and constants). The LLM's in-context learning and crossover capabilities (Meyerson et al., 2023) are then employed to refine the suggested equation skeletons in an iterative process. By representing equations as Python programs, we take advantage of LLM's ability to generate structured and executable code (Li et al., 2023; Shojaee et al., 2023) while providing a flexible and effective way to represent general mathematical relations. The program representation also facilitates direct and differentiable parameter optimization to better optimize the coefficients or constants in the generated equations.

To leverage LLM's scientific prior knowledge yet prevent the risk of LLM recitation (Wu et al., 2023) in equation discovery (observed for common benchmarks like Feynman (Udrescu & Tegmark, 2020)), we designed four custom benchmark problems across physics, biology, and materials science for the evaluation of LLM-SR. By incorporating synthetic modifications to physical models and experimental datasets, these problems aim to simulate the real discovery processes (see App. C and D for details). We evaluated LLM-SR using GPT-3.5-turbo (Brown, 2020) and Mixtral-8x7B (Jiang et al., 2024) as backbone LLMs. Results demonstrate that LLM-SR consistently outperforms state-of-the-art symbolic regression methods, discovering physically accurate equations with better fit and generalization in both in-domain (ID) and out-of-domain (OOD) test settings. By leveraging the scientific prior knowledge, LLM-SR explores the equation search space more efficiently, requiring fewer iterations to find accurate equations. Our ablation analysis also highlights the crucial role of data-driven feedback, iterative refinement, and program representation in LLM-SR's performance. The major contributions of this work are as follows:

- We introduce LLM-SR, a novel framework that leverages domain-specific prior knowledge and code generation capabilities of LLMs combined with off-the-shelf optimizers and evolutionary search for data-driven scientific equation discovery.

- We create four benchmark problems spanning physics, biology, and materials science, designed to simulate real-world discovery and prevent LLM recitation risks for evaluation of LLM-SR.

- We show that LLM-SR outperforms state-of-the-art symbolic regression methods by navigating the equation search space more efficiently and discovering more accurate equations with better out-of-domain generalization.

- We demonstrate through a comprehensive ablation study that natural language problem descriptions, program representation, data-driven feedback, and iterative hypothesis refinement are all essential components for LLM-SR's success.

## 2 LLM-SR METHODOLOGY

### 2.1 PROBLEM FORMULATION

In the task of data-driven equation discovery, also known as symbolic regression (SR), the goal is to find a concise symbolic expression $\tilde{f}$ approximating an unknown function $f : \mathbb{R}^d \to \mathbb{R}$. Given a dataset $\mathcal{D} = \{(\mathbf{x}_i, y_i)\}_{i=1}^n$, SR methods seek to uncover the hidden mathematical relationship such that $\tilde{f}(\mathbf{x}_i) \approx y_i, \forall i$. The discovered equation should not only accurately fit the observed data points but also exhibit strong generalization capabilities to unseen data while maintaining interpretability.

Current SR methods typically represent equations using techniques such as expression trees (Cranmer, 2023), prefix sequences (Petersen et al., 2021; Biggio et al., 2021), or context-free grammars (Brence et al., 2021). These representations provide structured and constrained search spaces, enabling evolutionary algorithms like genetic programming to explore and find candidate expressions. In contrast, we employ program functions to directly map inputs $\boldsymbol{x}$ to targets $\boldsymbol{y}$: `def` f(x): ... `return` y. This approach offers greater expressiveness in mathematical relations but expands the search space significantly. To navigate this vast program space effectively, we leverage LLMs for their scientific knowledge and code generation capabilities. Let $\pi_\theta$ denote a pre-trained LLM with parameters $\theta$. We iteratively sample equation program skeletons $\mathcal{F} = \{f : f \sim \pi_\theta\}$, aiming to maximize the reward $\text{Score}_\mathcal{T}(f, \mathcal{D})$ for a given scientific problem $\mathcal{T}$ and dataset $\mathcal{D}$: $f^* = \arg\max_f \mathbb{E}_{d \in \mathcal{D}} [\text{Score}_\mathcal{T}(f, \mathcal{D})]$. Our approach, LLM-SR, prompts the LLM to propose hypotheses based on problem specifications and demonstrations of previously discovered promising equations. The LLM generates equation program skeletons with placeholder parameters, which are then optimized using robust Python optimizers. Promising hypotheses are added to a dynamic experience buffer, guiding subsequent in-context example updates and equation refinement. Below we explain the key components of this framework, shown in Fig. 1, in more detail.

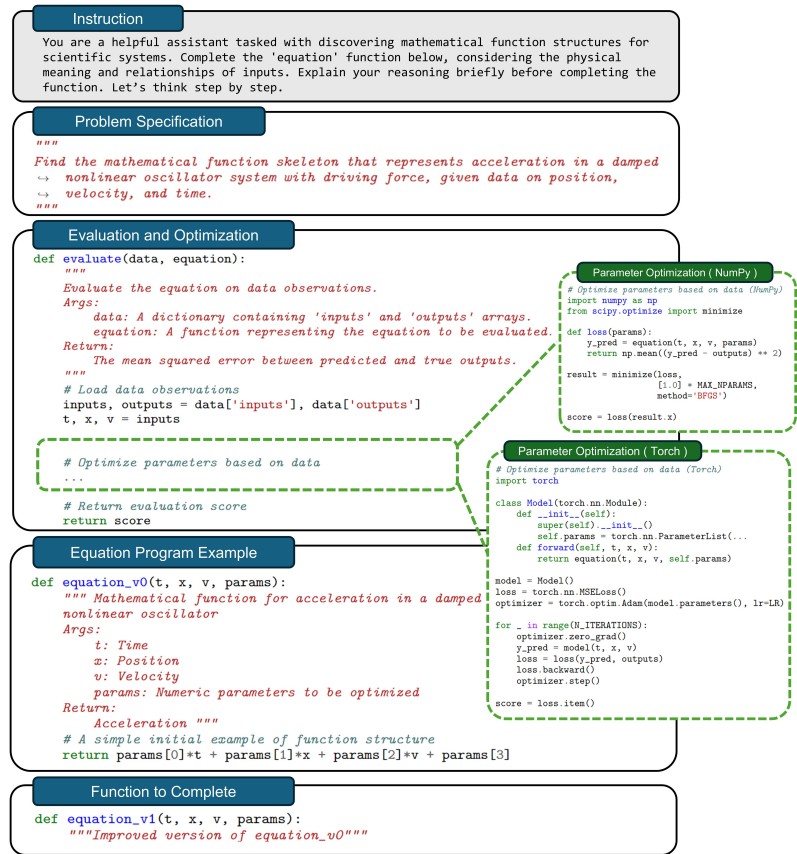

Figure 2: Example of initial input prompt for the nonlinear oscillator discovery task, including problem specification, evaluation and optimization function, and the initial input equation example.

## 2.2 HYPOTHESIS GENERATION

The hypothesis generation step (Fig. 1(a)) utilizes a pre-trained LLM to propose diverse and promising equation program skeletons. Our prompt structure, shown in Fig. 2, consists of the following components: **Instruction**: A clear directive for completing the function body, emphasizing consideration of physical meanings and relationships among input variables; **Problem Specification**: A concise description of the scientific problem, including key variables, constraints, and objectives; **Evaluation and Optimization Function**: The function used to assess the data-driven quality and fitness of proposed equations after parameter optimization; and **Experience Demonstration**: In-context examples of equation skeleton programs and their improvement trajectory.

At each iteration $t$, we sample a batch of $b$ equation skeletons $\mathcal{F}_t = \{f_i\}_{i=1}^b$ from the LLM $\pi_\theta$: $f_i \sim \pi_\theta(\cdot|\mathbf{p}_t)$ where $\mathbf{p}_t$ is the constructed prompt. We employ stochastic temperature-based sampling to balance exploration (creativity) and exploitation (prior knowledge) in the hypothesis space. Sampled equation programs are executed, and those failing to execute or exceeding a maximum execution time threshold are discarded to ensure validity and computational efficiency.

## 2.3 HYPOTHESIS OPTIMIZATION AND ASSESSMENT

After generating equation skeleton hypotheses, we evaluate and score them using observed data (Fig. 1(b)). This process involves optimizing the parameters of each hypothesis and then assessing its fitness. We decouple equation discovery into two steps: ($i$) discovering the equation program structures (skeletons) using the LLM, and ($ii$) optimizing the skeleton parameters/coefficients based on data. The LLM is responsible for generating equation skeletons and the core logic of the program, while the numeric values of the parameters are represented as placeholders in the form of a parameter vector `params` (as shown in Fig. 2). These placeholders are subsequently optimized to fit the data. Each equation program skeleton $f \in \mathcal{F}_t$ is a function of the form: "`def f(x, params): ... return y`". We employ two optimization approaches: **numpy+BFGS**: A nonlinear optimization method using `scipy` library (Fletcher, 1987), and **torch+Adam**: A stochastic gradient-based

optimization algorithm using `PyTorch` (Kingma & Ba, 2014). The choice between these methods depends on the problem characteristics and equation skeleton complexity. The numpy+BFGS is preferred for problems with fewer parameters, while torch+Adam is more suitable for larger-scale problems benefiting from efficient gradient computation through differentiable programming.

After optimizing the skeleton parameters (`params*`), we assess the fitness of equation program hypotheses by measuring its ability to capture underlying patterns in the data. We compute predicted target values as: $\hat{\mathbf{y}} = f(\mathbf{x}, \texttt{params}^*)$. The fitness evaluation score $s$ is then calculated as the negative Mean Squared Error (MSE) between predicted and true target values: $s = \text{Score}_{\mathcal{T}}(f, \mathcal{D}) = -\text{MSE}(\hat{\mathbf{y}}, \mathbf{y})$.

## 2.4 EXPERIENCE MANAGEMENT

To efficiently navigate the search landscape and avoid local minima, LLM-SR employs an experience management step (Fig.1(c)). This process maintains a diverse population of high-quality equation programs in a dynamic experience buffer and samples from this population to construct informative prompts for subsequent LLM iterations. Let $\mathcal{P}_t$ denote the experience buffer at iteration $t$, storing pairs of equation skeleton hypotheses and their corresponding scores $(f, s)$. We adopt an islands model (Cranmer, 2023; Romera-Paredes et al., 2024) with $m$ independently evolving islands, initialized with a copy of the equation program example from the initial prompt (`equation_v0` in Fig. 2). At each iteration $t$, new hypotheses $\mathcal{F}_t$ and their scores are added to the source island (from which the in-context examples of prompts were sampled) if they improve upon the current best: $\mathcal{P}_t^i \leftarrow \mathcal{P}_t^i \cup \{(f, s) : f \in \mathcal{F}_t, s = -\text{Score}_{\mathcal{T}}(f, \mathcal{D}), s > s_{\text{best}}^i\}$ where $\mathcal{P}_t^i$ is the $i$-th island and $s_{\text{best}}^i$ is its current best score. Within each island, equation programs are clustered based on their signature (defined by their score) to further preserve diversity.

To construct informative prompts, we then sample equation programs from the experience buffer using a two-stage method. First, uniformly select a random island from the $m$ available. Second, sample $k$ equation programs from the selected island using (a) Cluster selection via Boltzmann sampling, favoring higher scores: $P_i = \frac{\exp(s_i/\tau_c)}{\sum_{i'} \exp(s_{i'}/\tau_c)}$ where $s_i$ is the mean score of the $i$-th cluster and $\tau_c$ is a temperature parameter. (b) Individual program sampling, favoring shorter programs: $P(f_i) \propto \exp(-\tilde{l}_i/\tau_p)$ where $\tilde{l}_i$ is the normalized program length and $\tau_p$ is a temperature parameter. The sampled programs are then included in the prompt as in-context experience demonstrations, guiding the LLM in generating new equation program hypotheses. Detailed sampling procedures are provided in Appendix B.

Algorithm 1 presents the simplified pseudo-code of the LLM-SR framework. The experience buffer $\mathcal{P}_0$ is initialized with initial prompt, using a simple linear equation skeleton as a template (e.g., Fig. 2 for the nonlinear oscillator problem). This initial structure serves as a baseline for the LLM to modify operators and structures based on its domain knowledge. Each iteration $t$ involves: $(i)$ sampling $k$ in-context examples from $\mathcal{P}_{t-1}$, $(ii)$ updating the prompt, $(iii)$ generating $b$ equation program skeletons from the LLM, and $(iv)$ evaluating and potentially adding these to $\mathcal{P}_t$ if they improve upon the best score $s^*$. This process leverages the LLM's generative capabilities to refine equation structures guided by the evolving experience buffer. The algorithm returns the best-scoring program $f^*$ and its score $s^*$ as the optimal solution, iteratively exploring the equation space while balancing exploitation of promising structures with exploration of new possibilities.

## 3 EXPERIMENTS

### 3.1 BENCHMARKS AND DATASETS

The Feynman benchmark (Udrescu & Tegmark, 2020), comprising 120 fundamental physics problems from *Feynman Lectures on Physics database series*[1], is the current

---

**Algorithm 1:** LLM-SR

**Input** : LLM $\pi_\theta$, dataset $\mathcal{D}$, problem $\mathcal{T}$, $T$ iterations, $k$ in-context examples, $b$ samples per prompt

\# Initialize population
$\mathcal{P}_0 \leftarrow \text{InitPop}()$
$f^*, s^* \leftarrow \text{null}, -\infty$
**for** $t \leftarrow 1$ **to** $T-1$ **do**
   \# Sample examples from buffer
   $E \leftarrow \{e_j\}_{j=1}^k$,
   $e_j = \text{SampleExp}(\mathcal{P}_{t-1})$
   \# Prompt with new examples
   $\mathbf{p} \leftarrow \text{MakeFewShotPrompt}(E)$
   \# Sample from LLM
   $\mathcal{F}_t \leftarrow \{f_j\}_{j=1}^b, f_j \sim \pi_\theta(\cdot|\mathbf{p})$
   \# Evaluation and population update
   **for** $f \in \mathcal{F}_t$ **do**
      $s \leftarrow \text{Score}_{\mathcal{T}}(f, \mathcal{D})$
      **if** $s > s^*$ **then**
         $f^*, s^* \leftarrow f, s$
         $\mathcal{P}_t \leftarrow \mathcal{P}_{t-1} \cup \{(f, s)\}$
      **end**
   **end**
**end**

**Output** : $f^*, s^*$

---

[1] https://space.mit.edu/home/tegmark/aifeynman.html

standard benchmark for evaluating symbolic regression techniques in scientific equation discovery. However, our investigation reveals that LLMs have significant memorization issues with these well-known physics equations, potentially undermining their effectiveness in assessing LLM-based equation discovery approaches. For instance, LLM-SR rapidly achieves low data-driven errors within few iterations ($< 20$) on Feynman problems, suggesting a recitation of memorized information rather than a process of discovery (full results in App. C). To address these limitations and provide a more robust evaluation, we introduce novel benchmark problems across three scientific domains. Our benchmark design focuses on: ($i$) Custom modifications to physical models to prevent trivial memorization; ($ii$) Complex synthetic non-linear relationships to challenge creative exploration; and ($iii$) Realistic scenarios with experimental data to reflect real modeling processes. These benchmarks are designed to challenge the model's ability to uncover complex mathematical relations while leveraging its scientific prior knowledge, simulating conditions for scientific discovery. To validate our new benchmarks' effectiveness, we compared LLM response perplexity (using open-source Mixtral-8x7B) and equation discovery error curves (using GPT-3.5) between Feynman problems and our new benchmarks. Results show lower perplexity (Fig. 9 in App. C) and sharper discovery curves (Fig. 11 in App. C) for Feynman problems, suggesting that both LLM backbones have more likely memorized common Feynman equations, while our benchmarks present novel challenges requiring reasoning and exploration. We next discuss these new benchmark problems in detail:

**Nonlinear Oscillators**  Nonlinear damped oscillators, ubiquitous in physics and engineering, are governed by differential equations describing the complex interplay between an oscillator's position, velocity, and acting forces. The general form of these equations is typically expressed as: $\ddot{x} + f(t, x, \dot{x}) = 0$ where $t$ is time, $x$ is position, and $f(t, x, \dot{x})$ represents nonlinear forces. To challenge LLM-based equation discovery methods beyond common oscillator systems (e.g., Van der Pol, Duffing), we introduce two custom nonlinear designs: ① `Oscillation 1:` $\dot{v} = F \sin(\omega x) - \alpha v^3 - \beta x^3 - \gamma x v - x \cos(x)$; and ② `Oscillation 2:` $\dot{v} = F \sin(\omega t) - \alpha v^3 - \beta x v - \delta x \exp(\gamma x)$, where $v = \dot{x}$ represents velocity, and $\omega, \alpha, \beta, \gamma, \delta$ are constants. These two forms, serving as a proof of concept, are carefully designed to incorporate a combination of challenging yet solvable nonlinear structures (including trigonometric, polynomial, and exponential) that are distinct from well-known oscillator systems. More details on the design rationale and data generation are provided in App. D.1.

**Bacterial Growth**  The growth of Escherichia coli (E. coli) bacteria has been widely studied in microbiology due to its importance in various applications, such as biotechnology, and food safety. Discovering equations governing E. coli growth rate under different conditions is crucial for predicting and optimizing bacterial growth. The bacterial population growth rate has been modeled using a differential equation with the effects of population density ($B$), substrate concentration ($S$), temperature ($T$), and pH level, which is commonly formulated with multiplicative structure: $\frac{dB}{dt} = f(B, S, T, \text{pH}) = f_B(B) \cdot f_S(S) \cdot f_T(T) \cdot f_{\text{pH}}(\text{pH})$. To create a challenging benchmark that leverages LLMs' prior knowledge while preventing trivial memorization, we introduce novel nonlinear formulations for $f_T(T)$ and $f_{\text{pH}}(\text{pH})$. These custom functions maintain key characteristics of established models while introducing complexities that require exploration and discovery rather than recall. The complete mathematical formulations, along with the data generation process and parameter ranges, are detailed in App. D.2.

**Material Stress Behavior**  The stress-strain relationship of materials under varying conditions, particularly as a function of temperature and material type, is fundamental to structural design and analysis across engineering disciplines. This benchmark problem leverages a real-world experimental dataset from (Aakash et al., 2019), comprising tensile tests on Aluminum 6061-T651 across a range of temperatures. The inclusion of this benchmark serves multiple purposes: ($i$) It challenges LLM-based equation discovery methods with experimental data, moving beyond synthetic or idealized problems. ($ii$) Unlike the previous benchmarks, there is no predetermined theoretical model structure for this problem, necessitating creative modeling approaches from LLMs. In other words, modeling for this type of task is mostly empirical and the stress-strain-temperature relations may vary significantly based on the specific material and experimental condition, preventing trivial memorization. More details on this problem and experimental data are provided in App. D.3.

## 3.2 EXPERIMENTAL SETUP

We compare LLM-SR against state-of-the-art symbolic regression (SR) methods, including evolutionary-based approaches like **GPlearn**[2] (Genetic Programming) and **PySR**[3] (multi-island

---

[2]`https://gplearn.readthedocs.io/en/stable/`
[3]`https://github.com/MilesCranmer/PySR`

| Model | Oscillation 1 | | Oscillation 2 | | E. coli growth | | Stress-Strain | |
|---|---|---|---|---|---|---|---|---|
| | ID↓ | OOD↓ | ID↓ | OOD↓ | ID↓ | OOD↓ | ID↓ | OOD↓ |
| GPlearn | 0.0155 | 0.5567 | 0.7551 | 3.188 | 1.081 | 1.039 | 0.1063 | 0.4091 |
| NeSymReS (Biggio et al., 2021) | 0.0047 | 0.5377 | 0.2488 | 0.6472 | N/A ($d > 3$) | | 0.7928 | 0.6377 |
| E2E (Kamienny et al., 2022) | 0.0082 | 0.3722 | 0.1401 | 0.1911 | 0.6321 | 1.4467 | 0.2262 | 0.5867 |
| DSR (Petersen et al., 2021) | 0.0087 | 0.2454 | 0.0580 | 0.1945 | 0.9451 | 2.4291 | 0.3326 | 1.108 |
| uDSR (Landajuela et al., 2022) | 0.0003 | 0.0007 | 0.0032 | 0.0015 | 0.3322 | 5.4584 | 0.0502 | 0.1761 |
| PySR (Cranmer, 2023) | 0.0009 | 0.3106 | 0.0002 | 0.0098 | 0.0376 | 1.0141 | 0.0331 | 0.1304 |
| LLM-SR (`Mixtral`) | **7.89e-8** | **0.0002** | 0.0030 | 0.0291 | **0.0026** | **0.0037** | **0.0162** | 0.0946 |
| LLM-SR (`GPT-3.5`) | 4.65e-7 | 0.0005 | **2.12e-7** | **3.81e-5** | 0.0214 | 0.0264 | 0.0210 | **0.0516** |

Table 1: **Quantitative performance comparison** of LLM-SR (with GPT-3.5 and Mixtral backbones), and SR baseline models on different scientific benchmark problems measured by Normalized Mean Squared Error. "N/A" refers to incompatibility of E. coli Growth dataset for the NeSymReS baseline (limited to $< 3d$ data).

asynchronous evolution) (Cranmer, 2023), and deep learning-based methods such as **DSR** (reinforcement learning for expression generation) (Petersen et al., 2021) and **uDSR** (extending DSR with Genetic Programming search at decoding) (Landajuela et al., 2022). We also evaluate against pre-trained Transformer SR models: **NeSymReS** (Biggio et al., 2021) and **E2E** (Kamienny et al., 2022). This selection provides a comprehensive evaluation across different SR paradigms. We allow all search-based baselines to run for over 2M iterations until convergence to their best performance. In LLM-SR experiments, each iteration samples $b = 4$ equation skeletons per prompt with temperature $\tau = 0.8$, optimizes parameters via numpy+BFGS or torch+Adam (with 30 seconds timeout), and uses $k = 2$ in-context examples from the experience buffer for refinement. We run LLM-SR variants for around 2.5K iterations in all experiments. More details on the implementation and parameter settings of each baseline as well as implementation specifics of LLM-SR, including experience buffer structure, prompt refinement strategy, and parallel evaluation are provided in App. A.

### 3.3 QUANTITATIVE RESULTS

**Accuracy** Table 1 compares the performance of LLM-SR (using GPT-3.5 and Mixtral backbones) against state-of-the-art symbolic regression methods across various scientific benchmarks. Performance is measured using Normalized Mean Squared Error (NMSE), with lower values indicating better performance. LLM-SR with both backbones consistently outperform baselines, despite running for fewer iterations (2.5K vs. 2M+ for baselines). To assess generalization capability, we evaluate performance on both in-domain (ID) and out-of-domain (OOD) test sets. The performance gap between LLM-SR and baselines is more pronounced in the OOD setting, suggesting superior generalization of LLM-SR's discovered equations. For instance, on the E. coli growth problem, LLM-SR achieves an OOD NMSE of $\sim 0.0037$, significantly outperforming other methods (with OOD NMSE $> 1$).

Among baselines, PySR and uDSR show the best performance, while Transformer SR models (NeSymReS, E2E) perform poorly, likely due to limited generalization from their pretraining on common benchmark distributions to our novel datasets. These results demonstrate LLM-SR's effectiveness in discovering accurate and generalizable equations across diverse scientific domains.

**Efficiency** Fig. 3 shows the performance trajectories of LLM-SR variants and symbolic regression baselines across different scientific benchmark problems, depicting the best fitting scores achieved over search iterations. By leveraging scientific prior knowledge, LLM-SR explores a considerably lower number of equation candidates in the vast optimization space compared to symbolic regression baselines that lack this knowledge. This is evident from the sharp drops in

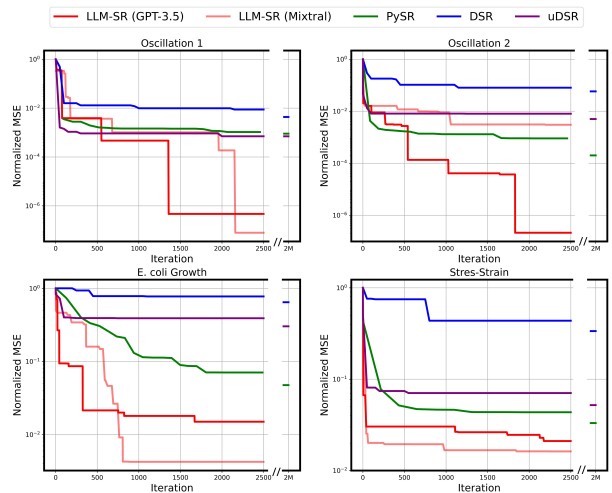

Figure 3: **Best score trajectories** of LLM-SR with GPT-3.5 and Mixtral against SR baselines across different benchmark problems. LLM-SR discovers accurate equations more efficiently, requiring fewer iterations. Baselines fail to match LLM-SR even after 2M iterations.

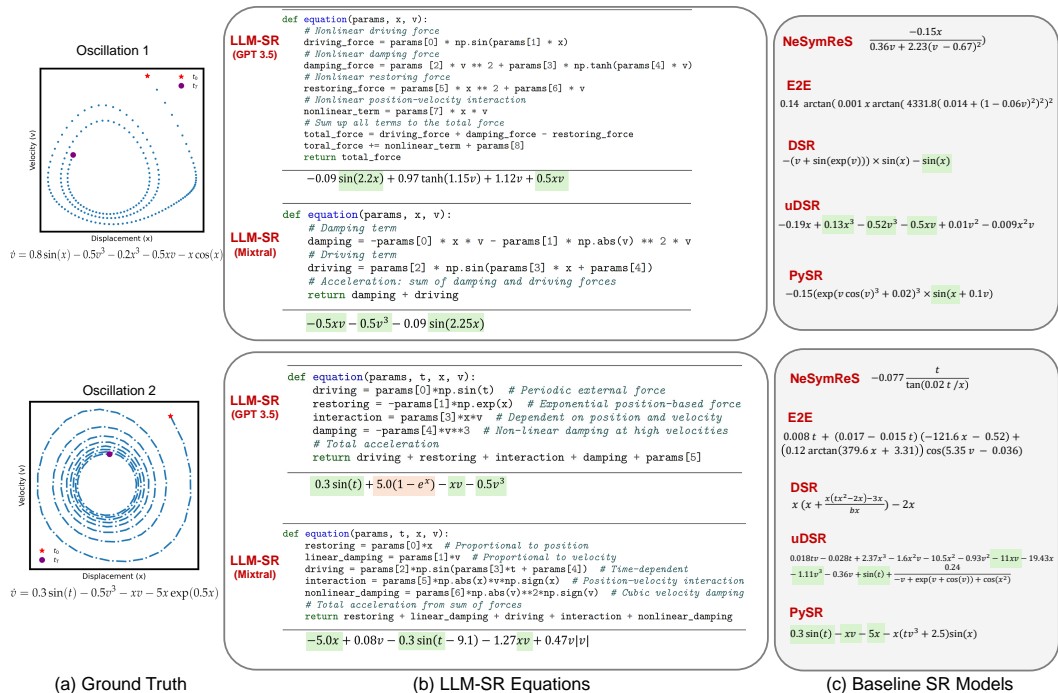

Figure 4: **Discovered equations for Oscillation 1 (top) and Oscillation 2 (bottom) problems**: **(a)** True equations and their phase diagram; **(b)** Equation program skeletons identified by LLM-SR, with simplified forms obtained after parameter optimization; and **(c)** Equations found using SR baselines. Shaded green terms denote recovered symbolic terms from true equations.

the error curves for LLM-SR variants, indicating they efficiently navigate the search space by exploiting domain knowledge to identify promising candidates more quickly. In contrast, the symbolic regression baselines show much more gradual improvements and fail to match LLM-SR's performance even after 2M+ iterations. The performance gap between LLM-SR and baselines also mostly widens over iterations, highlighting the effectiveness of LLMs acting as mutation (or crossover) agents in LLM-SR's iterative refinement process.

### 3.4 QUALITATIVE ANALYSIS

**Discovered Equations**  Fig. 4 presents the final discovered equations for both Oscillation problems using LLM-SR and other symbolic regression baselines. A notable observation is that equations discovered by LLM-SR have better recovered the symbolic terms of the true equations compared to baselines. Also, LLM-SR provides explanations and reasoning steps based on scientific knowledge about the problem, leading to more interpretable terms combined as the final function. For example, in both problems, LLM-SR identifies the equation structure as a combination of driving force, damping force, and restoring force terms, relating them to the problem's physical characteristics. In contrast, baselines generate equations lacking interpretability and understanding of the physical meanings of variables and the relations between them. These equations appear as a combination of mathematical operations and variables without a clear connection to the problem's underlying physical principles. App. G provides a more detailed qualitative analysis of the final discovered equations for other benchmark problems (Figs. 23 and 24), as well as the equations discovered over the performance trajectory of LLM-SR's iterations (Figs. 19-22).

**Generalization**  Fig. 5 compares predicted distributions obtained from LLM-SR, and competing baselines (PySR and uDSR) with the ground truth distribution of E. coli growth problem. The shaded region and black points indicate in-domain (ID) data, while the rest represent out-of-domain (OOD). Results show that distributions obtained from LLM-SR align well with the ground truth, not only for ID data but also for OOD regions. This alignment demonstrates the better generalizability of equations discovered by LLM-SR to unseen data, likely due to the integration of scientific prior knowledge in the equation discovery process. In contrast, PySR and uDSR tend to overfit the observed data, with significant deviations in OOD regions.

This overfitting behavior highlights their limited ability to generalize beyond the training data and capture the true physical underlying patterns. Detailed analyses for other benchmark problems are provided in App. G.

## 3.5 ABLATION STUDY

We conducted an ablation study on the Oscillation 2 problem using GPT-3.5 as the LLM backbone to investigate the impact of LLM-SR's key components (Fig. 6, more detailed results in App. E). Our findings reveal the crucial role of each component in the model's performance. The "*w/o Prior*" variant, which removes the natural language description of the scientific prob-

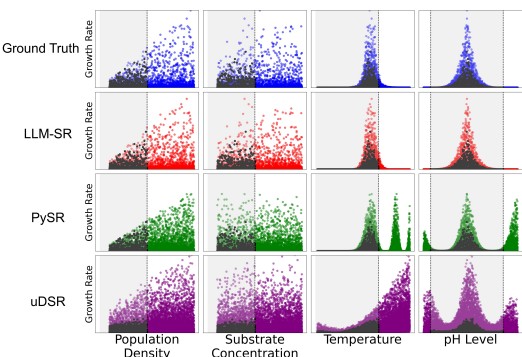

Figure 5: Comparison of E. coli growth rate distributions from LLM-SR, PySR, and uDSR.

lem and its variables, led to a considerable performance drop. This highlights the importance of incorporating prior domain knowledge in equation discovery. The "*w/o Program*" variant, which restricts LLM hypothesis generation to single-line mathematical expressions, also had a negative but less severe impact on performance, denoting the importance of programming flexibility in this task. The "*w/o Iterative Refinement*" variant, equivalent to the LLM sampling without the optimization loop, led to substantial performance drops (NMSE: 1.01e-1 in-domain, 1.81e-1 OOD), emphasizing the importance of the evolutionary search and optimization process in LLM-SR's success. The "*w/o skeleton + optimizer*" variant, which requires end-to-end equation generation without separate parameter optimization step (i.e., generating hypotheses as full equations along with their numeric parameters), also significantly worsened results (NMSE: 3.78e-1 in-domain, 3.75e-1 OOD). This highlights the effectiveness of our two-stage approach—generating equation skeletons followed by data-driven parameter optimization—in navigating complex combinatorial optimization space of discrete equation structures and continuous parameters.

We compared two optimization frameworks: **numpy+BFGS** and **torch+Adam**. In our experiments, the numpy+BFGS variant performed slightly better compared to torch+Adam. This difference is most likely attributed to the LLM's higher proficiency in generating numpy code rather than inherent superiority of the optimization method for this task. LLM-SR relies on direct and differentiable parameter optimization, a capability not present in current symbolic regression methods. Combining LLM-SR with LLM backbones that are better in generating PyTorch code could potentially enhance equation discovery by leveraging differentiable parameter optimization in future.

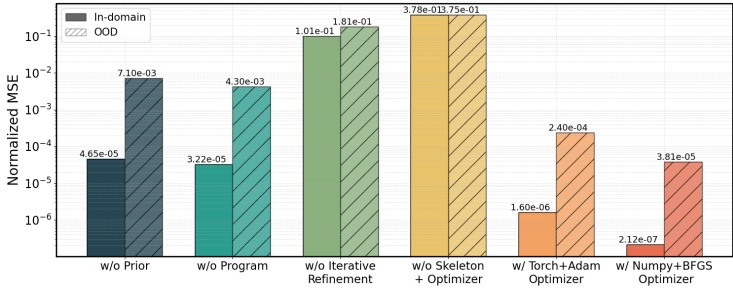

Figure 6: **Ablation results on the Oscillation 2 problem**, showing the impact of problem specification, program representation, iterative refinement, parameter optimization, and optimization frameworks on LLM-SR's performance.

## 4 RELATED WORK

**LLMs and Optimization** While LLMs have shown remarkable capabilities in various domains, their performance often falls short in tasks requiring high precision and complex reasoning. To address this, researchers have explored combining LLMs with feedback mechanisms (Madaan et al., 2024; Yang et al., 2023b; Haluptzok et al., 2022) and integrating them into iterative optimization loops (Lehman et al., 2023; Liu et al., 2023; Wu et al., 2024; Lange et al., 2024). Recently, LLMs have

been successfully applied in prompt optimization (Yang et al., 2023a; Guo et al., 2024), data-driven analysis (Majumder et al., 2024; Zheng et al., 2023b), and neural architecture search (Chen et al., 2023; Zheng et al., 2023a). Most related to our work is FunSearch (Romera-Paredes et al., 2024) that combines LLMs with systematic evaluators to search for heuristics that push the boundaries in solving some established open mathematical problems. Building upon these ideas, our LLM-SR framework employs LLM as an optimizer, leveraging its scientific prior knowledge and data-driven evaluators to discover mathematical equations underlying scientific observations.

**LLMs for Scientific Discovery** The integration of LLMs into scientific tasks has recently garnered significant attention, offering transformative potential across various fields such as drug discovery, biology, and materials science (Wang et al., 2023a; AI4Science & Quantum, 2023). Specifically, recent studies have demonstrated the capacity of LLMs to propose scientifically plausible and potentially novel hypotheses by leveraging their extensive domain knowledge and reasoning capabilities (Majumder et al., 2024; Zheng et al., 2023b; Qi et al., 2023; Ji et al., 2024). Also, when equipped with external tools and scientific simulators, LLM agents have shown promise in automated statistical discovery and reasoning (Li et al., 2024; Wang et al., 2023b; Ma et al., 2024). Despite the increasing exploration of LLMs in scientific contexts and question answering, their potential for tasks such as equation discovery and symbolic regression remains largely unexplored. Our work extends this line of research by introducing a novel approach for equation discovery that combines LLMs' scientific prior knowledge and code generation with data-driven evaluation.

**Symbolic Regression** Symbolic regression (SR) methods can be broadly categorized into search-based approaches, learning-based models, as well as hybrid learning and search methods. Search-based approaches mainly explore the space of equation structures and parameters using evolutionary algorithms or reinforcement learning (Schmidt & Lipson, 2009; Cranmer, 2023; Petersen et al., 2021; Sun et al., 2023). They offer interpretable results but often struggle with scalability and efficiency. Learning-based models, on the other hand, leverage large-scale synthetic data and Transformer models to learn the mapping between numeric input observations and output mathematical expressions (Biggio et al., 2021; Kamienny et al., 2022). Hybrid methods aim to combine the strengths of both approaches, guiding the search by employing neural priors to improve the expressiveness and efficiency of the discovery process (Landajuela et al., 2022; Shojaee et al., 2024; Mundhenk et al., 2021; Meidani et al., 2023). Despite the progress made by these approaches, they often face limitations such as the lack of scientific prior knowledge incorporation and the restricted expressiveness of traditional equation representations like expression trees. While there have been some works incorporating prior knowledge by using declarative bias and structures with pre-defined grammars (Todorovski & Dzeroski, 1997; Todorovski & Džeroski, 2007), these methods do not leverage the power of LLMs for this task. Our work advances this research direction by utilizing LLMs to efficiently search the combinatorial optimization space of equation discovery and generate meaningful equation structures based on the embedded scientific prior knowledge.

## 5 CONCLUSION AND FUTURE WORK

In this work, we introduced LLM-SR, a novel approach to equation discovery that leverages the scientific knowledge and code generation capabilities of Large Language Models (LLMs). By treating equations as programs and combining LLM-generated educated hypotheses with evolutionary search, our method demonstrates superior performance on benchmark problems across diverse scientific domains, particularly in out-of-domain test settings. Despite its promising results, LLM-SR has limitations. The method's performance is inherently tied to the quality and breadth of the LLM's training data, which may lead to biases or gaps in certain scientific domains. Additionally, the computational cost of iterative LLM queries and parameter optimization could be prohibitive for large-scale problems. Future work could focus on integrating domain-specific LMs and retrieval-augmented learning techniques to enhance the relevance and accuracy of generated equations; and incorporating human domain experts in the pipeline to improve the scientific plausibility. The creation of more comprehensive benchmarks, designed to simulate true discovery processes and prevent LLM recitation, is also crucial for rigorous evaluation of LLM-based equation discovery methods.

### ACKNOWLEDGMENTS

This research was partially supported by the U.S. National Science Foundation (NSF) under Grant No. 2416728.

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

APPENDIX

## A    BASELINE IMPLEMENTATION DETAILS

### A.1    MODELS

We compare LLM-SR against several state-of-the-art Symbolic Regression (SR) baselines, encompassing a diverse range of methodologies from traditional evolutionary approaches to modern deep learning-based techniques. The baselines include:

**GPlearn**    GPlearn is a pioneering and standard genetic programming (GP) SR approach. We use the open-source `gplearn`[4] package with the following parameters: Population size: 500, Tournament size: 20, Maximum generations: 2 million. Most of the hyperparameters are set from default setting.

**PySR**    PySR (Cranmer, 2023) is an advanced SR method that employs asynchronous multi-island GP-based evolutions. We implement PySR using the open-source `pysr`[5] package with the following settings: Number of populations: 15, Population size: 33, Maximum iterations: 2 million. Except for the number of iterations, other parameters are the same as the default setting in PySR. This configuration leverages the power of parallel evolution over a long time allowing for a diverse and robust search of the equation space.

**Deep Symbolic Regression (DSR)**    DSR (Petersen et al., 2021) employs an RNN-based reinforcement learning search over symbolic expressions. We implement DSR using the open-source `deep-symbolic-optimization (DSO)`[6] package with standard default parameters: Learning rate: 0.0005, Batch size: 512, and Maximum iterations: 2 million. This approach allows for a guided search through the space of symbolic expressions, leveraging the power of deep learning to inform the exploration process.

**Unified Deep Symbolic Regression (uDSR)**    uDSR (Landajuela et al., 2022) extends DSR by incorporating additional linear token and GP search at the decoding stage. We also implement uDSR using the `DSO` package with the same default parameters as DSR. This unified approach aims to combine the strengths of deep learning and traditional GP methods.

**Neural Symbolic Regression that Scales (NeSymReS)**    NeSymReS (Biggio et al., 2021) is the pioneering pre-trained Transformer SR model for expression skeleton generation. We implement it using the `NeuralSymbolicRegressionThatScales`[7] repository with the following default parameters: Number of datapoints passed to Transformer: 500, and Expression sampling size: 32. It is important to note that this model is limited to pre-training with $\leq 3$ variables. Consequently, we only apply the model to datasets with $d_{max} = 3$, excluding the Bacterial Growth problem (which has 4 variables) for evaluation of this model.

**End-to-End Symbolic Regression (E2E)**    E2E (Kamienny et al., 2022) is a more recent end-to-end pre-trained Transformer SR approach. We implement it using the `symbolicregression`[8] Facebook repository with the following default parameters: Number of datapoints passed to Transformer: 200, and Expression sampling size: 10. This model is also pre-trained for problems with $\leq 10$ variables.

### A.2    DATA PREPROCESSING AND MODEL EXECUTION

For the pre-trained Transformer SR models (NeSymReS and E2E), data normalization is crucial. We apply standard normalization to the input data before feeding it to these Transformer models to ensure optimal performance. For the search-based methods, we allow all baselines (GPlearn, PySR, DSR, and uDSR) to run for over 2 million iterations until convergence to their best performance. In the experiments, each baseline undergoes 5 replications. The best results obtained were then documented and reported. This extensive evaluation process, with a large number of iterations and search evaluations, ensures a robust assessment of each model's capability to converge towards optimal solutions and effectively explore the vast equation space of each problem.

---

[4] `https://gplearn.readthedocs.io/en/stable/`
[5] `https://github.com/MilesCranmer/PySR`
[6] `https://github.com/dso-org/deep-symbolic-optimization`
[7] `https://github.com/SymposiumOrganization/NeuralSymbolicRegressionThatScales`
[8] `https://github.com/facebookresearch/symbolicregression`

```
"""
Find the mathematical function skeleton that represents E. Coli bacterial growth
↪   rate, given data on population density of bacterial species, substrate
↪   concentration, temperature, and pH level.
"""

def evaluate(data, equation):
    """
    Evaluate the equation on data observations.
    ...
    """
    ...

def equation_v0(B, S, T, pH, params):
    """ Mathematical function for bacterial growth rate
    Args:
        B: Population density
        S: Substrate concentration
        T: Temperature
        pH: pH level
        params: Numeric parameters to be optimized
    Return:
        Growth rate """
    # A simple initial example of function structure
    return params[0]*B + params[1]*S + params[2]*T + params[3]*pH + params[4]

def equation_v1(B, S, T, pH, params):
    """Improved version of equation_v0"""
```

(a) Bacterial Growth Rate

```
"""
Find the mathematical function skeleton that represents stress, given  data on
↪   strain and temperature in an Aluminium rod for both elastic and plastic regions
"""

def evaluate(data, equation):
    """
    Evaluate the equation on data observations.
    ...
    """
    ...

def equation_v0(e, T, params):
    """ Mathematical function for stress in Alluminium rod
    Args:
        e: Strain level
        T: Temperature
        params: Numeric parameters to be optimized
    Return:
        Stress level """
    # A simple initial example of function structure
    return params[0]*e + params[1]*T + params[2]

def equation_v1(e, T, params):
    """Improved version of equation_v0"""
```

(b) Material Behavior Analysis

Figure 7: Example of input prompts program body for **(a)** E. Coli Growth and **(b)** Stress-Strain problems, with problem specification, and the initial equation program example (set as simple linear equation skeleton). For better readability, the details of evaluation function are not included in this figure. Check Fig. 2 for details.

## B    DETAILS OF LLM-SR METHOD AND IMPLEMENTATION

**Hypothesis Generation and Data-driven Evaluation**    Fig. 2 provided an example of specification for Nonlinear Oscillator problem. Here, Fig. 7 showcases illustrative examples of prompts and specifications tailored for the Bacterial Growth and Stress-Strain problems. These prompts contain descriptions of the problem and relevant variables, expressed in natural language. By providing this context, the language model can leverage its existing domain knowledge about the physical meaning and relations of variables to generate scientifically plausible hypotheses for new equation programs. Fig. 8 also shows a more detailed example of prompt and specification for LLM-SR that prompts the model to generate differentiable equation programs in `PyTorch` using tensor operations. The prompt suggests using differentiable operators and replacing non-differentiable components (e.g., if-else conditions) with smooth differentiable approximations.

Our experiments employ either Mixtral-8x7B (using 4 NVIDIA RTX 8000 GPUs with 48GB memory each) or GPT-3.5-turbo (via OpenAI API) as the language model backbone. During each prompting step, the language model generates $b = 4$ distinct equation program skeletons using a generation temperature of $\tau = 0.8$. This temperature setting is chosen based on preliminary experiments to

```python
"""
Find mathematical function form that fits data. This function form can
only contain binary and unary mathematical operators that are differentiable.
"""
import torch
import torch.nn as nn
import torch.optim as optim

class Model(torch.nn.Module):
    def __init__(self):
        super(Model, self).__init__()
        # Initialize model parameters
        self.params = torch.nn.ParameterList([torch.nn.Parameter(torch.tensor(1.0))
        ↪ for _ in range(P)])
    def forward(self, x: torch.Tensor) -> torch.Tensor:
        return
        # Foward pass the model
        equation(x, self.params)

def evaluate(data: dict) -> float:
    """
    Evaluate equation on input and output observations.
    """
    # Load true data observations
    inputs , outputs  = data['inputs'], data['outputs']
    # Define model
    model = Model()
    # Define optimizer
    optimizer = optim.Adam(model.parameters(), lr=0.001)
    # Optimize equation skeleton parameters
    model.train()
    num_iterations = 10000
    for i in range(num_iterations):
        # Zero the gradients
        optimizer.zero_grad()
        # Forward pass: compute predicted outputs by passing inputs to the model
        y_pred = model(inputs)
        # Compute the loss
        loss = torch.mean((y_pred - outputs) ** 2)
        # Backward pass: compute loss gradient with respect to parameters
        loss.backward()
        # Update parameters using optimizer
        optimizer.step()
    #Return evaluation score
    return -loss.item() if not (torch.isnan(loss) | torch.isinf(loss)).any() else
    ↪ None

def equation(x: torch.Tensor, params: torch.nn.ParameterList) -> torch.Tensor:
    """
    Args:
        x (torch.Tensor): Input data.
        params (torch.nn.ParameterList): List of model numeric constant parameters.
    Return:
        torch.Tensor: The result of applying the mathematical function to the x.
    """
    return params[0]*x + params[1]
```

Figure 8: An example of prompt structure, containing problem specification, evaluation and optimization function, and equation program with `pytorch` tensor operations.

balance creativity (exploration) and adherence to the problem constraints and reliance on the prior knowledge (exploitation). To control the length and the complexity of the generated equations and prevent overparameterization, we set the maximum number of parameters (length of `params` vector) as 10 in all experiments. The generated equation skeleton programs are then evaluated to gather feedback. In this framework, we deploy $e = 4$ evaluators to operate concurrently. This parallelization allows for rapid and efficient assessment of the generated programs per prompt. Evaluation is constrained by time and memory limits set at $T = 30$ seconds and $M = 2$GB, respectively. Equation programs that exceed these limits are disqualified and considered as discarded hypotheses by returning `None` scores. This constraint ensures timely progress and resource efficiency in the search process.

**Experience Buffer Management**  The system stores equation hypotheses and their corresponding data-driven scores in an experience buffer. It uses an islands model with multiple populations ($m = 10$ islands) to maintain diversity. Each island is initialized with a simple equation, which can be customized for domain-specific problems. At each iteration, new hypotheses and their fitness scores are added to their originating island if they improve upon the island's best score. To maintain the quality and diversity of the experience buffer, we follow (Romera-Paredes et al., 2024) and periodically reset the worst-performing islands. Every $T_{reset}$ iterations (every 4 hrs), we identify the $m/2$ islands whose best equation programs have the lowest fitness scores. All the equation programs in these islands are discarded, and each island is reinitialized with a single high-performing equation program, obtained by randomly selecting one of the surviving $m/2$ islands and copying its

highest-scoring equation program (favoring older programs in case of ties). This reset mechanism allows the framework to discard stagnant or unproductive regions of the equation program space and focus on more promising areas. Within each island, we further cluster the equation programs based on their signature, which is defined as the equation program score. Equation programs with identical signatures are grouped together, forming clusters within each island. This clustering approach helps preserve diversity by ensuring that equation programs with different performance characteristics are maintained in each population.

**Experience Sampling**  To construct informative prompts for the LLM, we sample equation programs from the experience buffer and update the prompt to include new experience demonstration in-context examples. Similar to (Romera-Paredes et al., 2024), here we use a two-stage sampling process. First, we randomly select an island from the $m$ available islands. Then, within the selected island, we sample $k$ equation programs (typically, $k = 2$) to be included as in-context examples in the prompt. When sampling equation programs within an island, we employ a two-step approach. First, we sample a cluster based on its evaluation score, favoring clusters with higher scores (i.e., higher-quality equation programs). Let $s_i$ denote the score of the $i$-th cluster, defined as an aggregation (e.g., mean) of all the scores in the signature that characterizes that cluster. The probability $P_i$ of choosing cluster $i$ is given by:

$$P_i = \frac{\exp\left(\frac{s_i}{\tau_c}\right)}{\sum_{i'} \exp\left(\frac{s_{i'}}{\tau_c}\right)}, \quad \tau_c = T_0 \left(1 - \frac{u \bmod N}{N}\right),$$

where $\tau_c$ is the temperature parameter, $u$ is the current number of equation programs in the island, and $T_0 = 0.1$ and $N = 10,000$ are hyperparameters. This selection approach is known as the Boltzmann selection procedure (Maza & Tidor, 1993). Once a cluster is selected, we sample an equation program within that cluster, favoring shorter programs. Let $l_i$ denote the negative length of the $i$-th program within the chosen cluster (measured as the number of characters), and let $\tilde{l}_i = \frac{l_i - \min_{i'} l_{i'}}{\max_{i'} l_{i'} + 10^{-6}}$. We set the probability of selecting each equation program proportional to $\exp\left(-\tilde{l}_i / \tau_p\right)$, where $\tau_p = 1$ is a temperature hyperparameter. The sampled programs are then included in the prompt as in-context experience demonstration, providing the LLM with relevant and diverse examples to guide the generation of new equation programs. By maintaining a diverse and high-quality population in the experience buffer and employing a strategic sampling approach, the experience management enables the LLM-SR framework to effectively explore the space of equation programs and iteratively refine its search based on the most promising candidates.

## C  LIMITATION OF FEYNMAN BENCHMARK PROBLEMS

The Feynman benchmark (Udrescu & Tegmark, 2020), consisting of 120 fundamental physics problems from the Feynman Lectures on Physics, is widely used to evaluate symbolic regression techniques in scientific equation discovery. However, our investigation indicates that LLMs have likely memorized many of these well-known physics equations. This memorization poses a challenge when using the Feynman benchmark to assess LLM-based equation discovery methods, as it may not accurately reflect the models' true discovery capabilities. This section elaborates on these limitations and provides evidence supporting the necessity of our newly designed benchmark problems.

**Perplexity Analysis**  To quantify the potential memorization of Feynman problems by LLMs, we first conducted a comparative perplexity analysis. Fig. 9 illustrates the median perplexity of Feynman problems against our new benchmarks (Oscillation 1, Oscillation 2, and E. Coli Growth) using the Mixtral-8x-7B model as the LLM backbone. Perplexity, in this context, is calculated only for the generation of equations given scientific context: $p(\text{Equation}|\text{Context})$. Examples of input prompts and outputs used for perplexity computation across different benchmarks are pro-

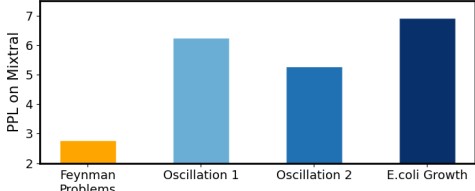

Figure 9: Perplexity (`Mixtral`) comparison of Feynman benchmark and our new designed benchmark problems

vided in Fig. 10. Mathematically, perplexity is defined as: $\text{PPL} = \exp\left(-\frac{1}{N}\sum_{i=1}^{N} \log p(x_i|x_{<i})\right)$, where $N$ is the number of tokens in the generated equation, and $p(x_i|x_{<i})$ is the probability of token

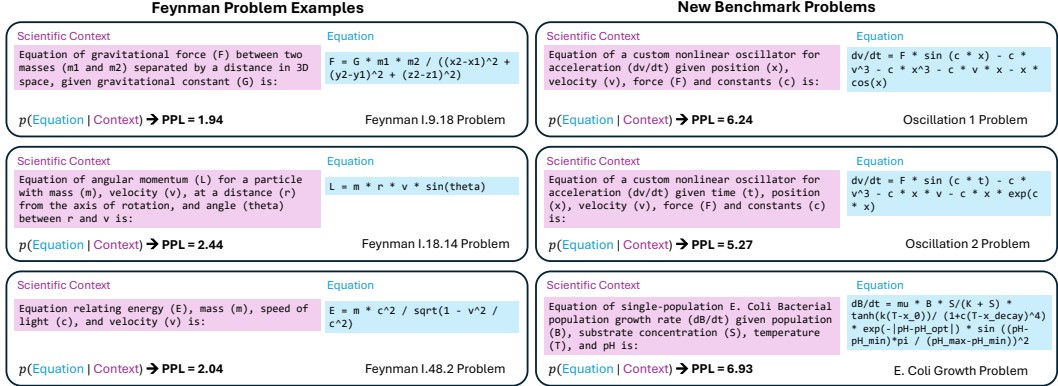

Figure 10: Examples of input context and output equations for Feynman equations and our new benchmark problems in Perplexity experiments with Mixtral LLM backbone.

$x_i$ given the preceding tokens, derived from the logits of the Mixtral model. The significantly lower perplexity observed for Feynman problems indicates a higher certainty in the LLM's predictions for these equations, and a higher chance of LLM recitation rather than reasoning and discovery. This suggests a high likelihood of memorization of well-known Feynman equations by the LLM, potentially due to their prevalence in scientific training data. It is worth noting that we have excluded the Stress-Strain problem from this analysis due to its experimental nature and lack of a predetermined theoretical model structure, precluding the calculation of perplexity.

**Discovery Error Curve Analysis** To further validate the need for new benchmarks, we compared equation discovery error curves between Feynman problems and our new benchmark problems. Fig. 11 presents the performance of LLM-SR with a GPT-3.5 backbone across various problems, showing the best score trajectory of Normalized Mean Squared Error (NMSE) against the number of iterations. For Feynman benchmark problems, LLM-SR achieves low NMSE scores within very few iterations, often in a single pass. This rapid convergence further supports the hypothesis that LLMs have likely memorized these fundamental physics equations due to their ubiquity in training data. Qualitative examples in Figs. 12 and 13 also provide additional evidence. The LLM's one-pass responses to several Feynman problems not only demonstrate functional accuracy but also often recite the exact form of the corresponding physics expressions, suggesting direct recall rather than a

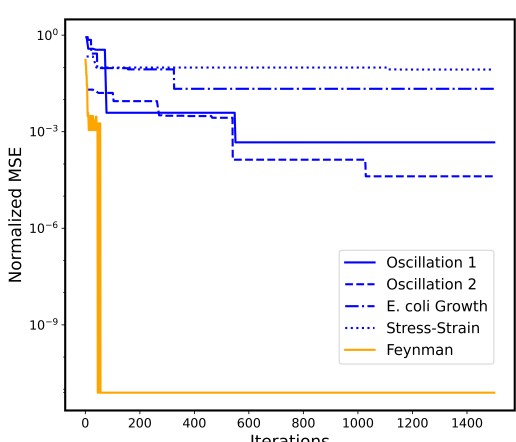

Figure 11: Trajectory of Normalized MSE score over iterations for LLM-SR (`GPT-3.5`) on Feynman benchmark problems versus new benchmarks

discovery process. In contrast, our newly designed benchmark problems present novel challenges requiring reasoning and exploration.

# D    ADDITIONAL DETAILS ON NEW BENCHMARK PROBLEMS

The datasets used in this study include both publicly available and newly generated data. The material stress behavior analysis dataset (stress-strain) is publicly available under the CC BY 4.0 license and can be accessed at https://data.mendeley.com/datasets/rd6jm9tyb6/1. The remaining datasets (Oscillation 1, Oscillation 2, and E. coli Growth) were generated for this work and are released under the MIT License as part of the LLM-SR GitHub repository: https://github.com/deep-symbolic-mathematics/LLM-SR

**Input Prompt**

```
"""
Find the mathematical function skeleton that represents the resultant intensity due
↪   to the interference of two coherent light sources.
"""

def evaluate(data, equation):
    """
    Evaluate the equation on data observations.
    ...
    """
    ...

def equation_v0(I1, I2, delta, params):
    """ Mathematical function for the resultant intensity due to interference of two
    ↪   waves
    Args:
        I1: Intensity of first wave
        I2: Intensity of second wave
        delta: Phase difference between two waves
        params: Numeric parameters to be optimized
    Return:
        Resultant intensity """
    # A simple initial example of function structure
    return params[0]*I1 + params[1]*I2 + params[2]*delta + params[3]

def equation_v1(I1, I2, delta, params):
    """Improved version of equation_v0"""
```

**LLM Response**

```
return params[0]*np.sqrt(I1*I2)*np.cos(delta) + params[1]*I1 + params[2]*I2 +
↪   params[3]
```

Feynman I.37.4 Problem 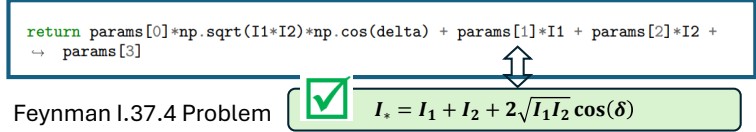 $I_* = I_1 + I_2 + 2\sqrt{I_1 I_2}\cos(\delta)$

Figure 12: An example of LLM response to Feynman I.37.4 problem, demonstrating LLM recitation without iterative search. For better readability, the details of evaluation function are not included in this figure. Check Fig. 2 for details.

## D.1 NONLINEAR OSCILLATOR EQUATIONS

In this work, we introduce two novel nonlinear oscillator systems as part of our effort to create more challenging and robust benchmarks for LLM-based equation discovery methods. These custom-designed oscillators extend beyond commonly studied systems like Van der Pol, Rayleigh, or Duffing oscillators, presenting unique challenges that test the reasoning and discovery capabilities of LLM-based approaches.

The general form of nonlinear damped oscillator equations is typically expressed as: $\ddot{x} + f(t, x, \dot{x}) = 0$, where $t$ represents time, $x$ represents position, and $f(t, x, \dot{x})$ represents nonlinear forces. Our custom designs expand upon this framework, incorporating a rich combination of nonlinear terms to create systems that are challenging yet solvable. We simulate two nonlinear oscillators using the `solve_ivp` function from the `scipy` library to generate data.

| Dataset | Time range | initial values | F | $\alpha$ | $\beta$ | $\delta$ | $\gamma$ | $\omega$ |
|---|---|---|---|---|---|---|---|---|
| Oscillator 1 | (0, 50) | {x=0.5, v=0.5} | 0.8 | 0.5 | 0.2 | - | 0.5 | 1.0 |
| Oscillator 2 | (0, 50) | {x=0.5, v=0.5} | 0.3 | 0.5 | 1.0 | 5.0 | 0.5 | 1.0 |

Table 2: Parameter values for Oscillator datasets.

The parameters and initial values for these simulations are provided in Table 2. The governing equations for our oscillator systems are as follows:

**Oscillator 1:**

$$\dot{v} = F\sin(\omega x) - \alpha v^3 - \beta x^3 - \gamma x \cdot v - x\cos(x)$$

**Oscillator 2:**

$$\dot{v} = F\sin(\omega t) - \alpha v^3 - \beta x \cdot v - \delta x \cdot \exp(\gamma x)$$

**Input Prompt**

```
"""
Find the mathematical function skeleton that represents the relationship between the
↪  angle of incidence and the angle of refraction.
"""

def evaluate(data, equation):
    """
    Evaluate the equation on data observations.
    ...
    """
    ...

def equation_v0(n, theta, params):
    """ Mathematical function for angle of refraction
    Args:
        n: Refractive index
        theta: Angle of incidence
        params: Numeric parameters to be optimized
    Return:
        Refraction angle """
    # A simple initial example of function structure
    return params[0]*n + params[1]*theta + params[2]

def equation_v1(n, theta, params):
    """Improved version of equation_v0"""
```

**LLM Response**

```
return params[0] * np.arcsin(params[1] * n * np.sin(theta)) + params[2]
```

Feynman I.26.2 Problem ✅ $\theta_1 = \arcsin(n \sin(\theta_2))$

Figure 13: An example of LLM response to Feynman I.26.2 problem, demonstrating LLM recitation without iterative search. For better readability, the details of evaluation function are not included in this figure. Check Fig. 2 for details.

where $v = \dot{x}$ represents velocity, and $F, \omega, \alpha, \beta, \gamma, \delta$ are constants specific to each oscillator system. These equations are carefully deisgned to incorporate a diverse set of nonlinear structures, including trigonometric, polynomial, and exponential terms. This design choice serves multiple purposes:

- **Challenging Complexity**: The combination of various nonlinear terms creates a rich dynamical system that is more complex than common oscillator systems, making the equation discovery task non-trivial.

- **Realistic Physics**: While complex, these equations are still solvable and still represent physically plausible systems, incorporating recognizable elements such as nonlinear damping and position-dependent restoring forces.

- **Novelty**: By deviating from well-known oscillator forms, we reduce the likelihood of LLMs simply reciting memorized equations, thus testing their reasoning and discovery capabilities in the context of data-driven scientific equation discovery.

Fig. 14 illustrates the phase plane diagrams of these nonlinear damped oscillators, visually demonstrating the complex dynamics arising from the interplay of nonlinear driving forces, restoring forces, and damping forces. These diagrams highlight the rich behavior that makes these systems challenging for equation identification tasks. To effectively evaluate the generalization capability of predicted equations, we employ a strategic data partitioning scheme. The simulation data is divided into three sets based on the trajectory time: (1) Training set, (2) In-domain validation set, and (3) Out-of-domain validation set. Specifically, we utilize the time interval $T = [0, 20]$ to evaluate the out-of-domain generalization of the discovered equations.

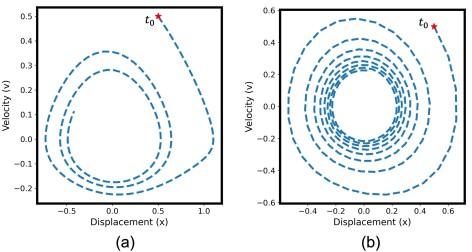

Figure 14: Phase diagrams of trajectories corresponding to custom oscillators: **(a)** Oscillator 1 and **(b)** Oscillator 2

## D.2 E. COLI GROWTH RATE EQUATIONS

In the domain of microbiology, understanding and modeling the growth dynamics of Escherichia coli (E. coli) is of paramount importance due to its wide-ranging applications in biotechnology, food safety, and fundamental biological research. To advance the LLM-based equation discovery approaches in this field, we have developed a novel benchmark problem centered around E. coli growth rate modeling. The growth rate of bacterial populations, including E. coli, is typically modeled by a differential equation that incorporates multiple environmental factors. This mathematical model commonly takes a multiplicative form:

$$\frac{dB}{dt} = f(B, S, T, \text{pH}) = f_B(B) \cdot f_S(S) \cdot f_T(T) \cdot f_{\text{pH}}(\text{pH}),$$

where $B$ represents bacterial population density, $S$ is substrate concentration, $T$ is temperature, and pH represents the acidity or alkalinity of the growth medium. To create a benchmark that is grounded in biological prior knowledge yet is challenging, we have extended this framework with a custom differential equation:

$$\frac{dB}{dt} = \mu_{max} B \left( \frac{S}{K_s + S} \right) \frac{\tanh\left(k(T - x_0)\right)}{1 + c(T - x_{decay})^4} \exp\left(-\left|\text{pH} - \text{pH}_{opt}\right|\right) \sin\left(\frac{(\text{pH} - \text{pH}_{min})\pi}{\text{pH}_{max} - \text{pH}_{min}}\right)^2$$

This equation incorporates several key components, each designed to test different aspects of equation discovery systems:

- **Population Density** ($f_B$): We maintain a linear relationship with $B$, reflecting a simple single-population scenario. This choice allows the focus to remain on the more complex environmental dependencies.

- **Substrate Concentration** ($f_S$): We employ the well-established Monod equation, $\left(\frac{S}{K_s + S}\right)$, which has been a cornerstone of bacterial growth modeling since its introduction by Jacques Monod in 1949. This inclusion serves as prior knowledge, allowing us to evaluate how well discovery methods can identify known relationships within a more complex overall structure.

- **Temperature Dependency** ($f_T$): We introduce a novel formulation, $\frac{\tanh(k(T - x_0))}{1 + c(T - x_{decay})^4}$, which captures the non-monotonic response of bacterial growth to temperature changes. This function combines a hyperbolic tangent term, representing the initial growth acceleration with temperature, and a quartic decay term, modeling the rapid decline in growth rate at high temperatures. This formulation presents a new and challenging form for LLM-based equation discovery methods, as it introduces operators and structures not commonly seen in the literature of this scientific context.

- **pH Dependency** ($f_{\text{pH}}$): Our custom pH function, $\exp\left(-\left|\text{pH} - \text{pH}_{opt}\right|\right) \sin\left(\frac{(\text{pH} - \text{pH}_{min})\pi}{\text{pH}_{max} - \text{pH}_{min}}\right)^2$, combines exponential and trigonometric terms to model the complex relationship between bacterial growth and pH levels. This formulation captures both the optimal pH range for growth and the symmetric decline in growth rate as pH deviates from the optimum. It also poses challenging setting for LLM-based equation discovery methods with structures uncommon in the relevant scientific context.

Fig. 15 illustrates the behavior of our custom-designed $f_T(T)$ and $f_{\text{pH}}(\text{pH})$ functions in comparison to established models from the literature for temperature and pH impact in bacterial growth. As observed, our custom models maintain key characteristics of bacterial growth responses while introducing complexities that challenge equation discovery methods. The temperature dependency model shows a sharper optimal peak and more rapid decline at high temperatures compared to traditional models, while the pH dependency model exhibits a narrower optimal range with steeper declines outside this range.

This carefully constructed benchmark serves multiple purposes: ($i$) It leverages LLMs' prior knowledge of bacterial growth patterns and common mathematical functions used in biological modeling; ($ii$) It prevents trivial LLM recitation by introducing novel combinations of functions and operators that go beyond standard models; and ($iii$) It challenges equation discovery systems to identify complex, biologically plausible relationships from data, simulating the process of scientific discovery.

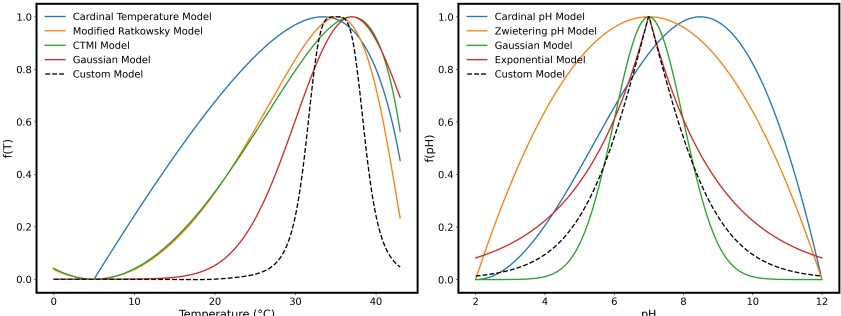

Figure 15: Scheme of some established models from literature for temperature and pH impact in bacterial growth compared to our custom-designed model behavior.

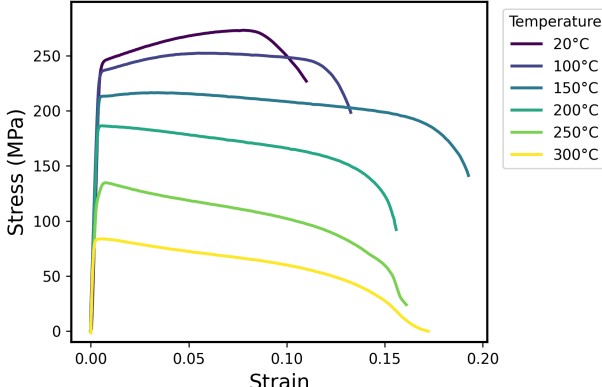

Figure 16: Stress-strain curves of Aluminium 6061-T651 under various temperatures (data from (Aakash et al., 2019))

### D.3   MATERIAL STRESS BEHAVIOR ANALYSIS

The analysis of material stress behavior with regard to temperature variations is a critical area of study in materials science and engineering. Our focus on experimental data of Aluminium 6061-T651 (Aakash et al., 2019) provides a new experimental case study for LLM-based equation discovery methods. Fig. 16 presents stress-strain curves for this alloy across a range of temperatures, offering rich insights into its mechanical behavior. For this problem, the experimental data represents the tensile behavior of material under uniaxial tension for 6 different temperatures from 20°C to 300°C. We allocate the data corresponding to $T = 200$°C for use as the out-of-domain validation set.

The stress-strain curves in Fig. 16 reveal several key features: **Temperature Dependence**: As temperature increases from 20°C to 300°C, we observe a significant decrease in both yield strength and ultimate tensile strength. **Elastic Region**: The initial linear portion of each curve represents the elastic region, where deformation is reversible. The slope of this region, known as Young's modulus, appears to decrease with increasing temperature, indicating reduced stiffness at higher temperatures. **Plastic Region**: Beyond the yield point, the curves exhibit non-linear behavior characteristic of plastic deformation. The shape of this region varies with temperature, suggesting changes in work hardening behavior. **Failure Region**: The endpoints of the curves indicate material failure. Notably, the strain at failure generally increases with temperature, implying enhanced ductility at higher temperatures. **Complex Non-linearity**: The overall shape of the curves exhibit piece-wise form, particularly in the plastic region, displays complex non-linear behavior that varies significantly with temperature and cannot be simply modeled with closed-form mathematical expressions.

The complex, nonlinear behavior of materials under varying experimental conditions poses significant challenges in developing comprehensive theoretical models. Moreover, unlike many physics problems where equation forms might be known or suspected, stress-strain-temperature relationships for specific materials often lack a universally accepted theoretical model. This absence necessitates

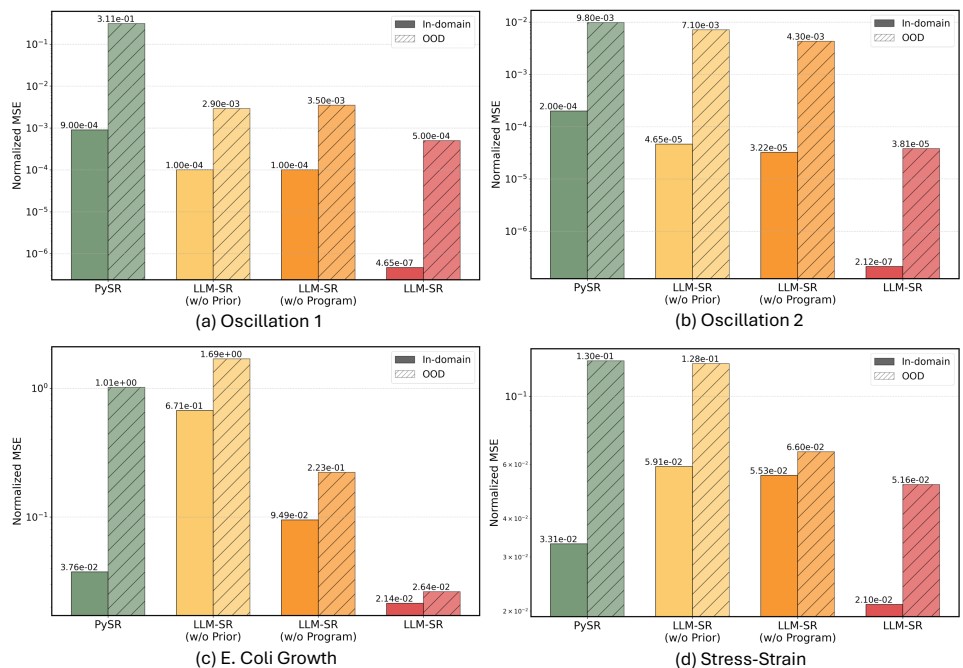

Figure 17: Ablation comparison of LLM-SR variants (`GPT-3.5` backbone) and PySR on four benchmarks, showing in-domain and out-of-domain errors for **(a)** Oscillation 1, **(b)** Oscillation 2, **(c)** E.coli Growth, and **(d)** Stress-Strain problems. LLM-SR consistently outperforms PySR and ablated versions, highlighting the importance of both prior knowledge and equation program representation components.

a more exploratory, data-driven approach to empirical equation discovery. By incorporating this real-world experimental materials science problem into our benchmark problems, we aim to evaluate the ability of LLM-SR to discover physically meaningful and interpretable equations in a domain where empirical modeling often dominates

## E    ADDITIONAL ABLATION STUDY

To evaluate the key components of our LLM-SR framework, we conducted additional ablation studies across all benchmark datasets. These experiments specifically target two main motivations behind LLM-SR: (1) leveraging scientific prior knowledge and (2) utilizing code generation capabilities of LLMs in the context of scientific equation discovery. We present the results of these ablation studies in Fig. 17, which compares the performance of three LLM-SR variant models: ($i$) LLM-SR (*w/o Prior*): This variant removes the incorporation of problem-specific prior knowledge from the LLMs' input prompt; ($ii$) LLM-SR (*w/o Program*): This variant eliminates the use of program representation for hypotheses; and ($iii$) LLM-SR: This variant is the final version of LLM-SR model including all components. We evaluated these models on all benchmark problems. For each problem, we assessed performance under both in-domain and out-of-domain (OOD) test settings, providing a robust evaluation of generalization capabilities in discovered equations.

Fig. 17 shows that both ablated components are critical to LLM-SR's success. Across all datasets and in both test settings, *w/o Prior* and *w/o Program* consistently underperformed compared to the full LLM-SR model. In both Oscillation 1 and Oscillation 2 problems, prior knowledge and code generation capabilities demonstrated comparable impacts on model performance. This held true for both in-domain and OOD settings, suggesting that both components contribute similarly to the model's understanding and generalization of oscillatory systems. For the E. coli Growth problem, we observed a more pronounced effect of prior knowledge. This indicates that domain-specific knowledge plays a particularly crucial role in modeling for this problem. Interestingly, while both components had similar impacts on in-domain performance in stress-strain problem, prior knowledge appeared to have a more substantial effect on OOD performance. This aligns with our intuition that OOD generalization should correlate more strongly with prior domain knowledge of the system.

| Model | Stress-Strain | | Oscillation 2 | |
|---|---|---|---|---|
| | ID↓ | OOD↓ | ID↓ | OOD↓ |
| LLM-SR (w/o multi-island & sampling) | 0.0257 | 0.1010 | 6.23e-6 | 0.0008 |
| LLM-SR | **0.0210** | **0.0516** | **2.12e-7** | **3.81e-5** |

Table 3: Impact of multi-island design on LLM-SR performance (with GPT-3.5 backbone) across two benchmark problems measured by Normalized Mean Squared Error.

We further conducted ablation experiments to analyze the impact of the multi-island buffer design and corresponding sampling strategy. Table 3 compares the performance of LLM-SR with and without the multi-island components on two benchmark problems. Specifically, we evaluated a variant that uses only one island in the buffer and employs a simple deterministic top-k selection approach for in-context example selection. The results demonstrate that the multi-island design positively impacts LLM-SR performance in both in-domain and OOD settings. Qualitatively, we observe that the number of islands plays a critical role in balancing exploitation and exploration. With fewer islands, the framework exhibits reduced exploration capabilities, generating less diverse equation hypotheses and converging prematurely to equation structures produced in early iterations.

## F    ADDITIONAL EXPERIMENTS

### F.1    NOISE ROBUSTNESS ANALYSIS

To evaluate the robustness of LLM-SR to imperfect data conditions commonly encountered in real-world settings, we conducted a systematic analysis of model performance under varying levels of noise. We specifically focused on the Oscillation 2 benchmark problem, introducing controlled Gaussian noise with different standard deviations ($\sigma = \{0, 0.01, 0.05, 0.1\}$) to the training data. This analysis provides insights into how the incorporation of prior knowledge through LLM-SR affects equation discovery performance when dealing with noisy measurements.

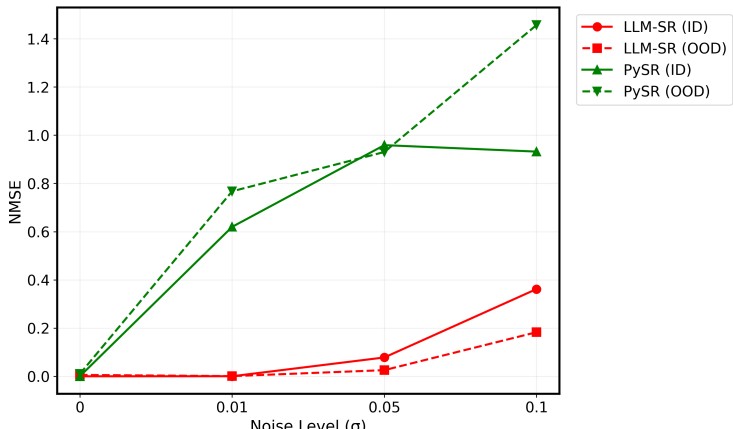

Figure 18: Noise robustness analysis of LLM-SR compared to PySR on the Oscillation 2 benchmark. The plot shows normalized mean squared error (NMSE) for both in-domain and out-of-domain (OOD) predictions under different levels of Gaussian noise ($\sigma = 0, 0.01, 0.05, 0.1$). While performance degrades with increasing noise for both methods, LLM-SR maintains better robustness, particularly for OOD predictions, demonstrating the value of incorporating domain knowledge in noisy real-world settings.

Fig. 18 presents a comparative analysis between LLM-SR and PySR under different noise conditions. The results demonstrate that while increasing noise levels generally degrade performance across all methods, LLM-SR exhibits notably better resilience to noise compared to traditional approaches. Specifically, at moderate noise levels ($\sigma = 0.01$, $\sigma = 0.05$), LLM-SR maintains significantly lower NMSE compared to PySR, particularly in out-of-domain predictions.

This enhanced robustness can be attributed to the incorporation of domain knowledge through LLM prompting, which helps constrain the search space to physically plausible solutions even in the presence of noise. These findings highlight an important aspect of symbolic regression: as data

quality decreases, the value of incorporating prior knowledge becomes increasingly significant. This observation aligns with the broader principle that when evidence (data) becomes less reliable, the role of priors in inference becomes more crucial.

## F.2 LLM-BASED OPTIMIZATION BASELINES

Several frameworks have emerged recently exploring the integration of LLMs into optimization tasks (Meyerson et al., 2023; Yang et al., 2023a; Romera-Paredes et al., 2024). Among these works, LMX (Meyerson et al., 2023) has included symbolic regression as one of its experimental tasks, though primarily as a proof-of-concept rather than aiming to achieve state-of-the-art performance. Their implementation does not incorporate domain-specific prior knowledge, generates complete equations with LLM instead of optimizable skeletons, uses older LLM models (Galactica, Pythia), and lacks several design elements present in LLM-SR such as equation-as-program representation and multi-island dynamic memory management for diverse exploration. We conducted experiments to evaluate LLM-SR alongside LMX and FunSearch (Romera-Paredes et al., 2024). For LMX, we utilized their open-source implementation directly, while for FunSearch, we adapted their prompt and feedback design to suit the equation-as-program task. All methods were run for the same number of iterations as LLM-SR, using GPT-3.5 as the LLM backbone.

| Model | Oscillation 1 | | Oscillation 2 | |
|---|---|---|---|---|
| | ID↓ | OOD↓ | ID↓ | OOD↓ |
| DSR | 0.0087 | 0.2454 | 0.0580 | 0.1945 |
| uDSR | 0.0003 | 0.0007 | 0.0032 | 0.0015 |
| PySR | 0.0009 | 0.3106 | 0.0002 | 0.0098 |
| LMX | 0.5031 | 48.93 | 1.004 | 0.9371 |
| FunSearch | 0.4840 | 8.059 | 0.7234 | 0.5861 |
| LLM-SR (w/o skeleton+optimizer) | 0.1371 | 0.6764 | 0.3780 | 0.3751 |
| LLM-SR (w/o Prior) | 0.0001 | 0.0029 | 4.65e-5 | 0.0071 |
| LLM-SR (w/o Program) | 0.0001 | 0.0035 | 3.22e-5 | 0.0043 |
| LLM-SR | **4.65e-7** | **0.0005** | **2.12e-7** | **3.81e-5** |

Table 4: Additional experimental results comparing different optimization methods on oscillator problems measured by Normalized Mean Squared Error.

Results in Table 4 show that LLM-SR and its variants achieve better performance than LMX and FunSearch in both in-domain and OOD test settings. Even when the skeleton+optimizer design is ablated (allowing LLM to generate complete equations without placeholder parameters), performance drops significantly but still outperforms both LMX and FunSearch. Further ablation of prior knowledge or program representation components shows less dramatic impact, though the full LLM-SR incorporating all components achieves the best performance overall. Notably, traditional SR baselines (DSR, uDSR, PySR) also outperform the existing LLM-based approaches on both datasets, supporting our focus on comparing against state-of-the-art SR methods in the main experiments.

## F.3 LLM BACKBONES

We have also conducted experiments with larger LLMs as the backbone for LLM-SR (than current Mixtral and GPT-3.5 backbones). Due to the computational budget limitations, we only conducted experiments on the oscillation 1 and oscillation 2 datasets. For a fair comparison, we ran this GPT-4o backbone model for the same number of iterations (2500) as other LLM-SR runs. Table 5 shows that on both datasets, LLM-SR (with GPT-4o) provides slightly better performance than LLM-SR with GPT-3.5 and Mixtral backbones, particularly in the OOD settings (reported numbers are NMSE as in Table 1). These findings align with expectations that improved LLMs with better knowledge, reasoning, and programming capabilities have potential to enhance performance in LLM-SR framework.

| Model | Oscillation 1 | | Oscillation 2 | |
|---|---|---|---|---|
| | ID↓ | OOD↓ | ID↓ | OOD↓ |
| LLM-SR (`Mixtral`) | **7.89e-8** | 0.0002 | 0.0030 | 0.0291 |
| LLM-SR (`GPT-3.5`) | 4.65e-7 | 0.0005 | 2.12e-7 | 3.81e-5 |
| LLM-SR (`GPT-4o`) | 7.29e-6 | **6.75e-5** | **4.27e-10** | **1.29e-6** |

Table 5: Additional experimental results comparing different LLM backbones on oscillator problems measured by Normalized Mean Squared Error.

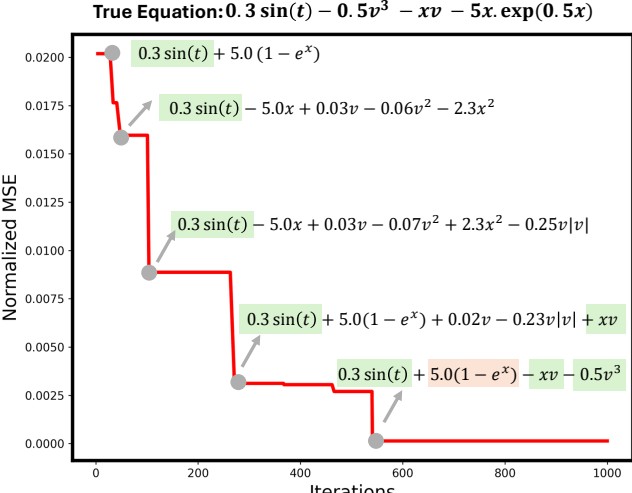

Figure 19: Performance Trajectory of LLM-SR (`GPT-3.5`) along with the best-scoring simplified equations (after parameter optimization) over iterations on the Oscillation 2 problem. Green highlights indicate recovered symbolic terms from the true equation.

## G ADDITIONAL QUALITATIVE RESULTS

**Discovery Trajectory** In this section, we evaluate the progress of generated equations using LLM-SR over iterations. This analysis can illustrate the qualitative evolutionary refinement of the discovered equations.

Fig. 19 shows the NMSE values for the Oscillation 2 dataset. For simplicity, we have provided the simplified equation versions of programs with their optimized parameters. We observe that some of the common nonlinear terms such as sinisoidal driving force term are found early in the search, while more complicated nonlinear terms are found later in the search. An interesting observation here is that while ground truth equation for this dataset is $\dot{v} = 0.3\sin(t) - 0.5v^3 - x \cdot v - 5x \cdot \exp(0.5x)$, LLM-SR has discovered the equation $\dot{v} = 0.3\sin(t) - 0.5v^3 - x \cdot v + 5(1 - \exp(x))$ at the end. By evaluating the different terms in these two forms, we observe that in fact $5(1 - \exp(x)) \approx -5x \cdot \exp(0.5x)$ for $x \in (-2, 2)$, which is the approximate range of displacement in this dataset.

Fig. 20 illustrates the evolution of equation program skeletons for the Oscillation 1 problem. It can be observed that the model attempts to incorporate various nonlinear terms corresponding to driving, restoring, and damping forces, as evidenced by comments or variable names within the code, aiming to enhance accuracy.

Similarly, Fig. 21 presents an annotated performance curve illustrating LLM-SR's performance on the E. coli growth rate equation discovery benchmark problem. It becomes apparent that the model recognizes the potential presence of optimal values for temperature and pH from the early iterations which comes from model prior knowledge about the common bell-shaped effect of these variables on the growth rate (check Fig. 15). To enhance accuracy, the model necessitates exploration and incorporation of various nonlinear forms. Notably, LLM-SR directs its evolutionary changes towards the more critical and variable aspects of the problem, specifically the pH and temperature effects, as opposed to other components such as substrate concentration represented by the Monod equation $\frac{S}{K+S}$. Additionally, the figure demonstrates LLM-SR's comprehension that different components of the function should be multiplied together in the final step, underscoring how prior domain knowledge of the problem structure can guide LLM-SR's evolutionary steps.

Fig. 22 displays three distinct equation skeleton programs discovered by LLM-SR for the stress-strain problem over search iterations. As in previous cases, we notice the model's enhancement through exploration and incorporation of various nonlinear terms into the equations, particularly in the plastic region terms. An additional significant observation for this problem is that stress-strain relationships often exhibit piece-wise behavior (as it can also be observed in Fig. 16), which closed-form expression

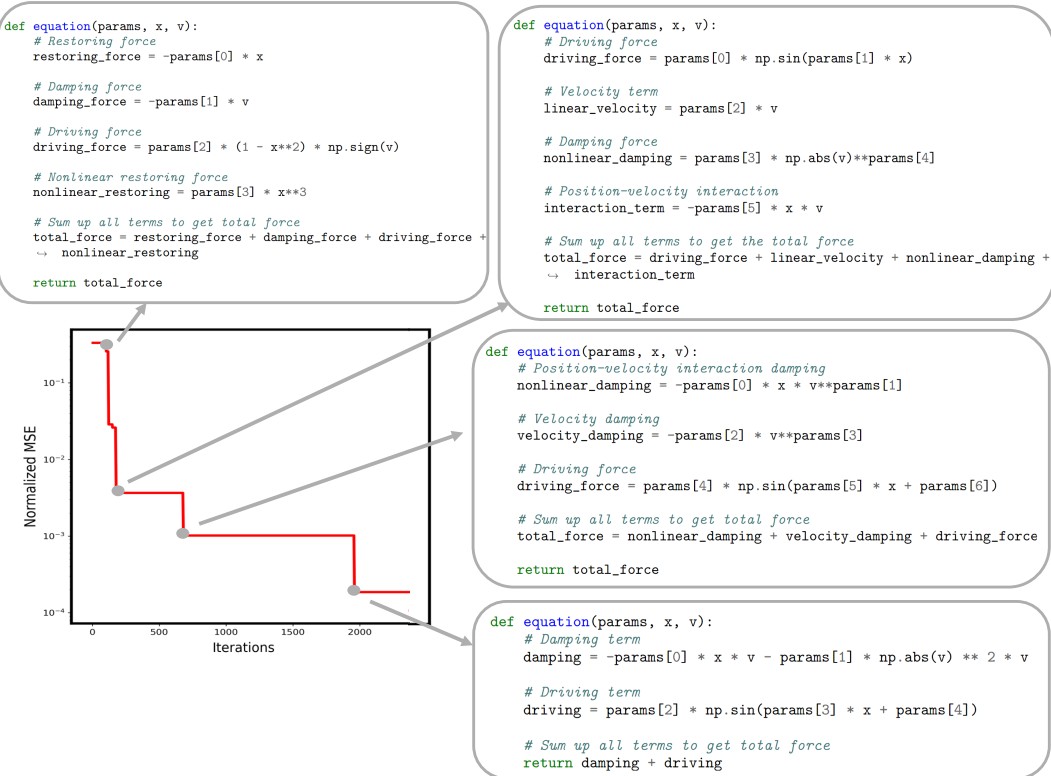

Figure 20: Performance Trajectory of LLM-SR (`Mixtral`) along with the best-scoring equation program skeletons (before parameter optimization) over iterations on the Oscillation 1 problem.

structures in traditional symbolic regression models struggle to model. However, LLM-SR represents equation skeletons as programs, thus, it can employ conditional rules (If-Else) or their continuous relaxations, utilizing step-wise nonlinear differentiable functions such as the `sigmoid` function to model smooth piece-wise relations. This differentiability and smooth approximation of if-else conditions are particularly helpful for the parameter optimization step, providing smooth functions for the optimizer to navigate.

**Discovered Equations**   Fig. 23 and Fig. 24 depict the equation programs identified by LLM-SR and other leading symbolic regression baselines (DSR, uDSR, and PySR) for the E. coli growth and the stress-strain problems, respectively. The diverse range of equation forms identified by different symbolic regression methods reflects the challenges posed by these datasets. Notably, in both datasets, the SR methods yield either lengthy or highly nonlinear equations that are not aligned with the prior knowledge of the systems, as evidenced by their poor out-of-domain (OOD) performance scores in Table 1. In contrast, LLM-SR finds flexible equation programs that are more interpretable and aligned with the domain-specific scientific priors of the systems.

**Behavior of Discovered Models**   Fig. 25 and Fig. 26 offer a qualitative comparison by visually presenting the outputs of the equations obtained using LLM-SR, PySR, and uDSR. Upon examination, it becomes evident that the predictions generated by LLM-SR exhibit a notable degree of alignment with the ground truth data. This alignment suggests that LLM-SR effectively captures the underlying patterns and dynamics of the data.

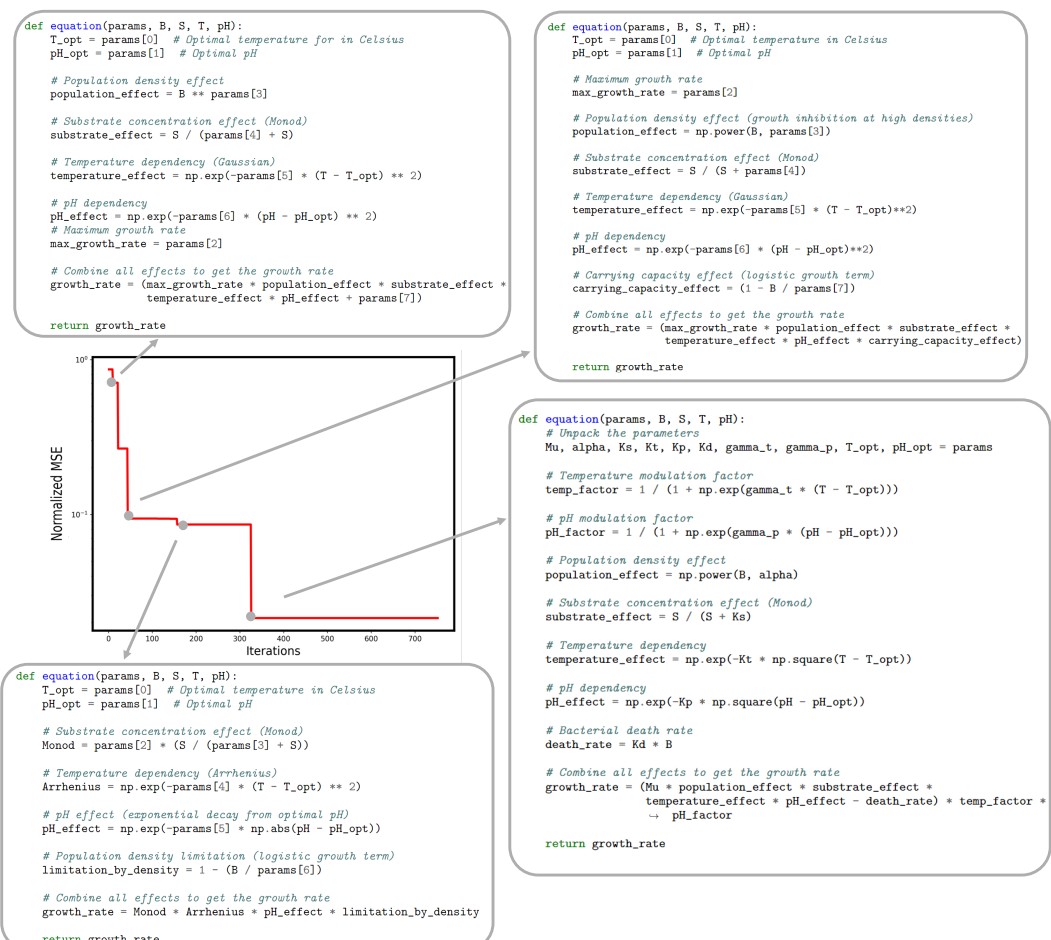

Figure 21: Performance Trajectory of LLM-SR (GPT-3.5) along with the best-scoring equation program skeletons (before parameter optimization) over iterations on the E. coli growth problem.

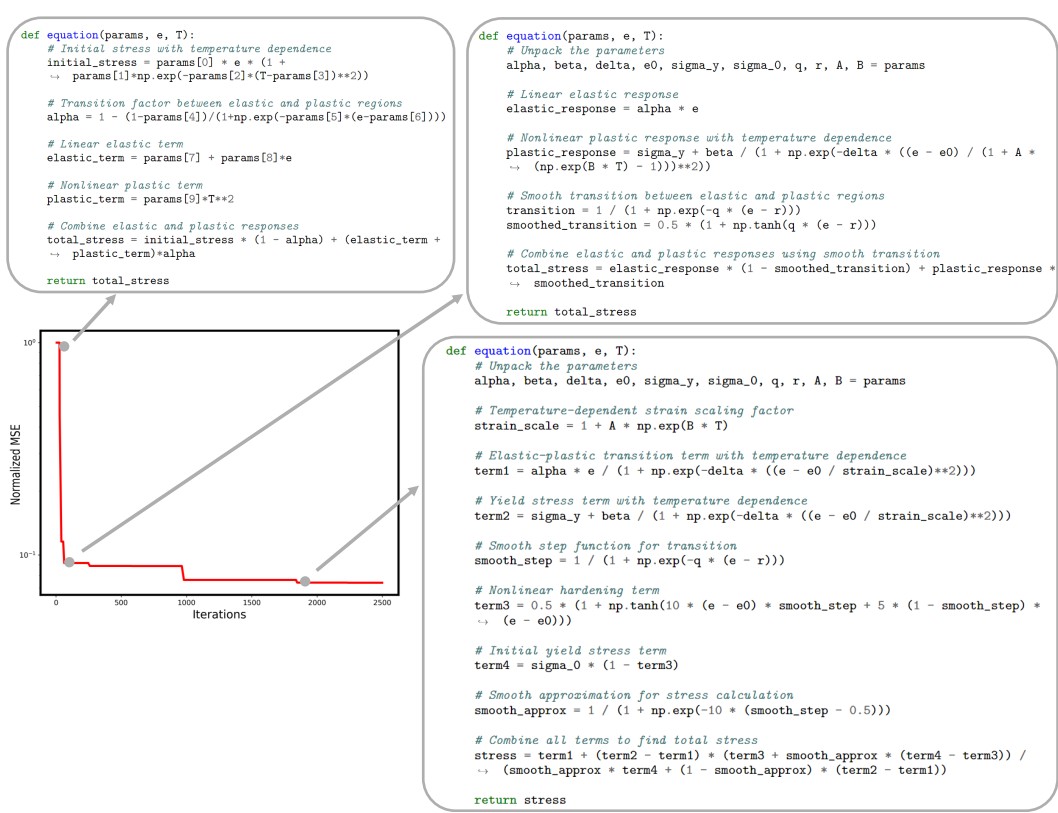

Figure 22: Performance Trajectory of LLM-SR (`Mixtral`) along with the best-scoring equation program skeletons (before parameter optimization) over iterations on the Stress-Strain problem.

```python
def equation(params, B, S, T, pH):                          # LLM-SR (Mixtral)
    # Unpack the parameters
    Mu, K, Ks, Kt, Kh, E, F, G, H, I, T_opt, pH_opt = params

    # Temperature factor
    temperature_factor = np.exp(-Kt * np.abs(T - T_opt) / E - ((T - T_opt) ** 2) /
    ↪  (2 * E ** 2))

    # pH factor
    pH_factor = 1 / (1 + Kh * np.exp(np.abs(pH - pH_opt) / F))

    # Inhibition factor
    inhibition_factor = 1 + I * np.clip(np.abs(pH - pH_opt) + np.exp(-np.abs(pH -
    ↪  pH_opt)), 0, 10)

    # Temperature-dependent growth factor
    h_factor = G * (T - T_opt) + H
    growth_factor = (1 + np.tanh(0.5 * (h_factor + np.arctan(h_factor)))) *
    ↪  pH_factor * inhibition_factor

    # Kinetic factor (Monod)
    kinetic_factor = S / (Ks + S)

    # Growth rate term
    growth_rate = Mu * B * np.where(
        kinetic_factor > 1,
        kinetic_factor / (1 + np.exp(-K * (kinetic_factor - 1))),
        kinetic_factor)

    # Combine all effects to get final growth rate
    growth_rate *= temperature_factor * growth_factor * np.maximum(1 - np.abs(T -
    ↪  T_opt) / 10, 0)

    return growth_rate
```

```python
def equation(params, B, S, T, pH):                          # LLM-SR (GPT 3.5)
    # Unpack the parameters
    Mu, alpha, Ks, Kt, Kp, Kd, gamma_t, gamma_p, T_opt, pH_opt = params

    # Temperature modulation factor
    temp_factor = 1 / (1 + np.exp(gamma_t * (T - T_opt)))

    # pH modulation factor
    pH_factor = 1 / (1 + np.exp(gamma_p * (pH - pH_opt)))

    # Population density effect
    population_effect = np.power(B, alpha)

    # Substrate concentration effect (Monod)
    substrate_effect = S / (S + Ks)

    # Temperature dependency
    temperature_effect = np.exp(-Kt * np.square(T - T_opt))

    # pH dependency
    pH_effect = np.exp(-Kp * np.square(pH - pH_opt))

    # Bacterial death rate
    death_rate = Kd * B

    # Combine all effects to get the growth rate
    growth_rate = (Mu * population_effect * substrate_effect *
                   temperature_effect * pH_effect - death_rate) * temp_factor *
                   ↪  pH_factor

    return growth_rate
```

$$\frac{T \log\left(\log(pH)\right)}{pH \times \left(b + s.pH + T \log(b)\right)} \qquad \text{DSR}$$

$$\begin{aligned}
&\Big(-9.1757 \times 10^{-5}B^3 + 3.29934 \times 10^{-4}B^2S + 1.5599 \times 10^{-5}B^2T - 3.96705 \times 10^{-5}B^2\text{pH} \\
&+ 6.62027 \times 10^{-5}B^2 + 4.40703 \times 10^{-4}BS^2 - 6.75274 \times 10^{-5}BST - 1.55073 \times 10^{-4}BS\text{pH} \\
&- 2.9761 \times 10^{-3}BS - 5.48518 \times 10^{-5}BT^2 - 1.27722 \times 10^{-6}BT\text{pH} + 1.40836 \times 10^{-3}BT \\
&+ 8.81533 \times 10^{-4}B\text{pH}^2 - 1.14007 \times 10^{-2}B\text{pH} + 3.00963 \times 10^{-2}B - 2.33351 \times 10^{-4}S^3 \\
&+ 3.01844 \times 10^{-5}S^2T - 1.31112 \times 10^{-4}S^2\text{pH} + 1.63188 \times 10^{-3}S^2 - 1.33597 \times 10^{-5}ST^2 \\
&- 1.7635 \times 10^{-5}ST\text{pH} + 4.1777 \times 10^{-4}ST + 2.98769 \times 10^{-4}S\text{pH}^2 - 2.5354 \times 10^{-3}S\text{pH} \\
&+ 3.0621 \times 10^{-3}S + 4.97662 \times 10^{-6}T^2 + 3.06327 \times 10^{-4}T\text{pH}^2 - 4.20817 \times 10^{-3}T\text{pH} \\
&+ 1.13549 \times 10^{-2}T - 1.23534 \times 10^{-4}\text{pH}^3 - 4.55259 \times 10^{-3}\text{pH}^2 + 7.83677 \times 10^{-2}\text{pH} - 0.231108\Big) \\
&\times \exp\left(\sin\left(\text{pH} + \frac{\text{pH}}{B + \text{pH}}\right)\right)
\end{aligned} \qquad \text{uDSR}$$

$$B \times \sin(3.5 \times 10^{-5} \times T^3)^6 \times \sin(\cos(\,|\sin(0.45pH)|+0.2))^6 \qquad \text{PySR}$$

Figure 23: Final discovered equations from LLM-SR and other leading SR baseline methods (DSR, uDSR, PySR for E. coli bacterial growth rate problem.

```python
def equation(params, e, T):                                    LLM-SR (Mixtral)
    # Unpack the parameters
    alpha, beta, delta, e0, sigma_y, sigma_0, q, r, A, B = params

    # Temperature-dependent strain scaling factor
    strain_scale = 1 + A * np.exp(B * T)

    # Elastic-plastic transition term with temperature dependence
    term1 = alpha * e / (1 + np.exp(-delta * ((e - e0 / strain_scale)**2)))

    # Yield stress term with temperature dependence
    term2 = sigma_y + beta / (1 + np.exp(-delta * ((e - e0 / strain_scale)**2)))

    # Smooth step function for transition
    smooth_step = 1 / (1 + np.exp(-q * (e - r)))

    # Nonlinear hardening term
    term3 = 0.5 * (1 + np.tanh(10 * (e - e0) * smooth_step + 5 * (1 - smooth_step) *
     ↪  (e - e0)))

    # Initial yield stress term
    term4 = sigma_0 * (1 - term3)

    # Smooth approximation for stress calculation
    smooth_approx = 1 / (1 + np.exp(-10 * (smooth_step - 0.5)))

    # Combine all terms to find total stress
    stress = term1 + (term2 - term1) * (term3 + smooth_approx * (term4 - term3)) /
     ↪  (smooth_approx * term4 + (1 - smooth_approx) * (term2 - term1))

    return stress
```

```python
def equation(params, e, T):                                    LLM-SR (GPT 3.5)
    # Unpack the parameters
    k1, k2, k3, k4, k5, k6, k7, k8, k9, k10 = params

    # Stress in the elastic region
    elastic_stress = k1 * e + k2 * T

    # Stress in the plastic region
    plastic_stress = k3 * (e ** 2) + k4 * T + k5

    #Stress in the failure region
    failure_stress = k6 * np.exp(k7 * e) + k8 * (T ** 2) + k9

    # Smooth transition from elastic to plastic region
    elastic_to_plastic = 1 / (1 + np.exp(-k8 * (e - k9)))

    # Smooth transition from plastic to failure region
    plastic_to_failure = 1 / (1 + np.exp(-k10 * (e - k9)))

    # Stresses in all regions
    total_stress = (
        elastic_stress * (1 - elastic_to_plastic) * (1 - plastic_to_failure) +
        plastic_stress * elastic_to_plastic * (1 - plastic_to_failure) +
        failure_stress * plastic_to_failure
    )

    return total_stress
```

$$\sin\left(\exp\left(-\epsilon \cdot T \cdot \exp\left(T^2\right) + \epsilon - T\right)\right) \qquad \textbf{DSR}$$

$$-3.51\epsilon^3 + 2.29\epsilon^2 \cdot T + 3.81\epsilon^2 - 0.92\epsilon \cdot T^2 - 0.30\epsilon \cdot T - 4.09\epsilon \qquad \textbf{uDSR}$$
$$+ 1.01T^3 - 2.01T^2 + 0.87T - \log\left(T - \log\left(\epsilon^2\right)\right) + 2.71$$

$$\sin\left(1.13\epsilon^2 \cdot \cos\left(\frac{T}{\cos(\epsilon)}\right)^2 \cdot \left(\frac{\epsilon}{\epsilon + 0.004}\right)^2\right) \qquad \textbf{PySR}$$

Figure 24: Final discovered equations from LLM-SR and leading SR baseline methods (DSR, uDSR, PySR) for Stress-Strain problem.

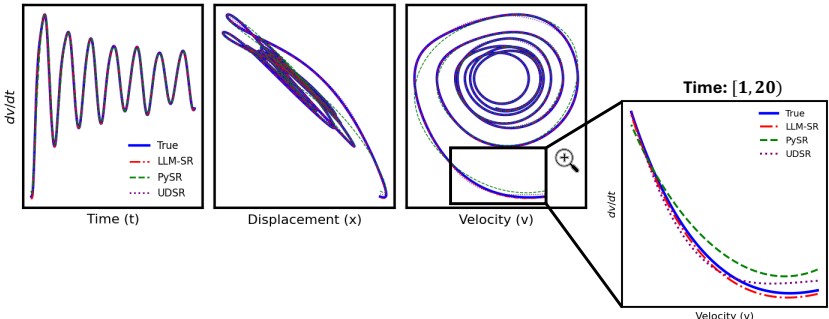

Figure 25: Qualitative evaluation of the performance of LLM-SR on Oscillation 2 problem compared to uDSR and PySR baselines. Plots show the target acceleration with respect to time, displacement, and velocity.

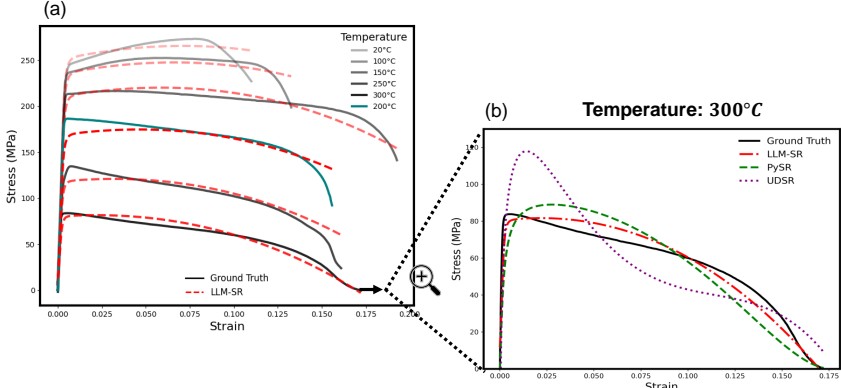

Figure 26: Qualitative evaluation of LLM-SR performance for Stress-Strain problem compared to uDSR and PySR baselines. Plots show the target stress with respect to strain and temperature.

