# OpenReview forum: "LLM-SR: Scientific Equation Discovery via Programming with Large Language Models"
_ICLR.cc/2025/Conference — ICLR 2025 Oral_

### Official Review · Reviewer_AFGC · 2024-11-02

**Soundness:** 3
**Presentation:** 3
**Contribution:** 3
**Rating:** 8
**Confidence:** 3

**Summary:**

The paper presents LLM-SR (Large Language Model Symbolic Regression), a framework that employs LLMs to discover scientific equations by treating equations as programs. LLM-SR combines LLMs with evolutionary search techniques to generate, evaluate, and refine equations that describe complex scientific phenomena. The process includes hypothesis generation by the LLM, parameter optimization of proposed equations, and a dynamic experience management system to refine hypotheses iteratively. The authors demonstrate LLM-SR’s superior ability to discover accurate and generalizable equations compared to traditional symbolic regression methods, especially in out-of-domain settings.

**Strengths:**

1. The approach of using LLMs to generate symbolic equation structures combines programming and scientific prior knowledge, setting it apart from traditional symbolic regression methods.
2. LLM-SR shows strong performance not only in-domain but also in out-of-domain scenarios, which is often challenging for symbolic regression techniques.
3. By leveraging scientific prior knowledge and a structured experience buffer, LLM-SR reduces the search space, achieving good results with fewer iterations compared to traditional methods.

**Weaknesses:**

1. The framework relies heavily on the embedded scientific knowledge of LLMs, which may be biased, incomplete, or inaccurate for certain scientific domains, potentially limiting its robustness.
2. Although the paper addresses different domains, it would benefit from more diverse and complex real-world datasets to further validate the method’s general applicability.

**Questions:**

1. How would LLM-SR perform with even larger LLMs or specialized scientific models?
2. How does LLM-SR perform with complex systems involving interactions across multiple equations?

---

> ### Author Response · Authors · 2024-11-20
> **Response to Reviewer AFGC**
>
> Thank you for your thoughtful comments. Please find our responses below.
>
> > * The framework relies heavily on the embedded scientific knowledge of LLMs, which may be biased, incomplete, or inaccurate for certain scientific domains, potentially limiting its robustness.
> >
>
> We agree that LLMs can hallucinate and scientific knowledge embedded in LLMs may be biased, incomplete, or inaccurate for certain domains. However, we emphasize that LLM-SR does not rely solely on this prior domain knowledge. Instead, it integrates this knowledge with data-driven insights by testing various equation structures on the given data. This combination of reasoning over prior knowledge and data-driven insights enhances the method's robustness against spurious information. Additionally, as noted in the paper (section 5, line 523-524), LLM-SR could further benefit from LLMs fine-tuned for specific target domains, or retrieval-augmented architectures that can provide more reliable domain-specific knowledge.
>
> ---
> > * How would LLM-SR perform with even larger LLMs or specialized scientific models?
> >
>
> In response to the reviewer's suggestion, we have conducted experiments with larger LLMs as the backbone for LLM-SR (than current Mixtral and GPT-3.5 backbones). Due to the limited time of rebuttal, we were only able to conduct experiments on the oscillation 1 and oscillation 2 datasets. For a fair comparison, we ran this GPT-4o backbone model for the same number of iterations (~2500) as other LLM-SR runs. Table below shows that on both datasets, LLM-SR (with GPT-4o) provides slightly better performance than LLM-SR with GPT-3.5 and Mixtral backbones, particularly in the OOD settings (reported numbers are NMSE as in Table 1). These findings align with expectations that improved LLMs with better knowledge, reasoning, and programming capabilities can enhance LLM-SR's performance.
>
>
> | Model        | Oscillation 1 (ID/OOD) | Oscillation 2 (ID/OOD) |
> |----------------------------|----------------------|----------------------|
> | LLM-SR (Mixtral)         | **7.89e-8** / 0.0002       | 0.0030 / 0.0291       |
> | LLM-SR (GPT-3.5)  | 4.65e-7 / 0.0005       | 2.12e-7 / 3.81e-5       |
> | LLM-SR (GPT-4o)  | 7.29e-6 / **6.75e-5**       | **4.27e-10** / **1.29e-6**       |
>
> ---
> > * Although the paper addresses different domains, it would benefit from more diverse and complex real-world datasets to further validate the method’s general applicability.
> >
> Thank you for the suggestion. We would like to highlight that in our current experiments, the stress-strain dataset already contains measurements from complex real experimental data (detailed in Appendix D.3), and LLM-SR performs better on this dataset compared to baselines (Table 1).
>
> In response to your comment, we also conducted experiments testing LLM-SR's sensitivity to dataset noise by applying varying levels of controlled Gaussian noise to the oscillation 2 benchmark problem. Our comparative analysis between LLM-SR and PySR (shown in blue in Appendix F and Figure 18) shows that while increased noise degrades performance across all methods, LLM-SR maintains better noise robustness than the baselines. Specifically, with moderate amounts of noise (σ = 0.01), LLM-SR maintains its performance (NMSE: 2.00e-4 in-domain, 7.00e-4 out-of-domain) while PySR's performance degrades substantially (NMSE: 6.20e-1 in-domain, 7.67e-1 out-of-domain), and this performance gap increases at higher noise levels (σ = 0.1). These observations on real-world and simulated data in various domains indicate the capabilities and the general applicability of LLM-SR. However, we agree with the reviewer that expanding the use of LLM-SR to more diverse real-world settings is indeed an important direction to further explore.
>
> ---
> > * How does LLM-SR perform with complex systems involving interactions across multiple equations?
> >
>
> Thank you for the thoughtful question. We would like to note that current SR baselines typically treat such problems by breaking them into independent equations (e.g., coupled ODE systems). To maintain a fair comparison with existing SR baselines, our experiments also focus on standard single-equation SR settings. While we have not explicitly tested LLM-SR on complex systems involving interactions across multiple equations, we believe its knowledge-guided hypothesis generation could offer advantages in such scenarios as well.  This remains an interesting direction for future exploration.

---

> > ### Comment · Reviewer_AFGC · 2024-11-25
> >
> > Thank you for your valuable responds！It solves my questions, and I will reconsider my score!

---

> > > ### Author Response · Authors · 2024-11-25
> > > **Thank you from Authors**
> > >
> > > Thank you once again for your time and valuable insights. We are glad that our rebuttal has resolved your concerns and appreciate the raised score.

---

> ### Author Response · Authors · 2024-11-22
> **Looking forward to discussion**
>
> Dear Reviewer AFGC,
>
> We hope our answers and new experiments have addressed your concerns and questions. Please let us know if you have any more questions before the end of the discussion period.
>
> Should there be no additional concerns, we kindly ask you to consider revising your score.
>
> Thank you for your time and thoughtful feedback!

---

### Official Review · Reviewer_BtD7 · 2024-11-04

**Soundness:** 3
**Presentation:** 3
**Contribution:** 3
**Rating:** 8
**Confidence:** 4

**Summary:**

This paper proposes a method for symbolic regression (SR) that combines LLM-based evolutionary search for program synthesis with numerical optimizers (such as BFGS and Adam) to tune the parameters of the generated programs. The authors, additionally, present four new SR problems that were created with synthetic modifications to address concerns regarding LLM memorization. Finally, ablations are provided to analyze the benefits of different design choices.

**Strengths:**

1. To the best of my knowledge, framing SR as equation _template_ discovery seems novel and a useful contribution.
2. The analyses presented to demonstrate LLM memorization of the Feynman dataset are quite convincing.
3. The new benchmark problems are well-motivated and look like sound contributions.
4. The paper is, overall, clearly written and easy to follow, and visualizations are used well in various places.

**Weaknesses:**

My main concerns with the work are twofold:
1. The paper reads in a manner that moderately overclaims novelty. The main idea appears to be the combination of numerical optimizers with LLM-based optimization over templates, besides the contribution of new SR problems. However, the motivation, currently, is primarily devoted to comparisons with non-LLM methods, which seems unfair. Since the framing of SR problems within the context of LLM-based optimization is not new (Table 1 from [1] is a useful reference), it would be more appropriate to motivate the work by describing existing LLM-based SR methods, clearly stating their problems, and then showing how the new method addresses them.
2. My second concern is regarding the choice of baselines. While I appreciate the inclusion of several standard SR methods, I think the comparisons do not precisely show the benefits of the paper's contributions.
  (a) Given prior works such as [1, 2, 3] in LLM-based SR/optimization, it would have provided utility to see the performance of the proposed method in comparison to its closest counterparts. Alternatively, if the argument is that some of the ablations subsume these methods, it would then be useful to see those ablations presented as the main baselines on all the datasets instead of only on a partial set.
  (b) The ablation results from Figure 6 show that a substantial drop in performance occurs when coefficient optimization is removed. Based on this, it would have been useful to see whether simply equipping the non-LLM baselines with a similar coefficient optimizer would close the gaps that we currently see in Table 1. I am currently not convinced that the LLM is needed to facilitate the evolutionary search of equation templates.

Minor:
1. The choice of describing the contents in Section 2.4 as "Experience Management" is mildly concerning. Given a large body of prior work in evolutionary computation, it would be appropriate to avoid fragmenting the literature for the benefit of differentiation and instead re-use existing terminology. If there is indeed an aspect that warrants a new term, please do correct me.
2. It was mildly odd to read that mathematical equations have been "unreasonably" effective in describing complex phenomena. Unclear why we think it is unreasonable.

[1] Meyerson et al., 2023. Language model crossover: Variation through few-shot prompting.
[2] Romera-Paredes et al., 2023. Mathematical discoveries from program search with large language models.
[3] Yang et al., 2024. Large Language Models as Optimizers.

**Questions:**

1. L243 briefly described how population initialization is performed. Is the linear equation skeleton manually written? How many skeletons are provided at initialization?
2. In L149, should the expectation be over the data points in $\mathcal{D}$?
3. In Appendix E, could you clarify what LLM-SR (w/o Prior) refers to? From the text described in the main paper, I could not find mention of the described "problem-specific prior knowledge" (L1220).

---

> ### Author Response · Authors · 2024-11-20
> **Response to Reviewer BtD7 (1)**
>
> Thank you for your insightful and constructive feedback on our paper. Please find our responses below.
>
> ---
> > * The paper reads in a manner that moderately overclaims novelty. The main idea appears to be the combination of numerical optimizers with LLM-based optimization over templates, besides the contribution of new SR problems. However, the motivation, currently, is primarily devoted to comparisons with non-LLM methods, which seems unfair. Since the framing of SR problems within the context of LLM-based optimization is not new (Table 1 from [1] is a useful reference), it would be more appropriate to motivate the work by describing existing LLM-based SR methods, clearly stating their problems, and then showing how the new method addresses them.
> >
>
> Thanks for the thoughtful feedback. While LLM-based optimization frameworks exist, we would like to clarify that their use for symbolic regression (SR) has been limited. Among the cited works, OPRO [3] focuses on prompt optimization, FunSearch [2] explores heuristic discovery for general optimization tasks like bin packing, and only LMX [1] has studied symbolic regression but with significant limitations. Notably, LMX framework is tested on multiple tasks, with SR being one of them. It approaches SR as a proof-of-concept rather than aiming to achieve state-of-the-art performance, and it struggles to outperform SR baseline methods even on relatively simple problems (see Figure 7 in [1]). LMX's approach to SR has several key limitations that undermine its performance on the task: (1) Lacks domain knowledge integration in the evolutionary search; (2) Does not leverage equation-as-program representation; (3) Generates complete equations with LLM instead of optimizable skeletons; (4) Lacks multi-island dynamic memory management for diverse exploration; and (5) Builds on the older LLM models (Galactica, Pythia) instead of more recent, capable ones.
>
> In light of these considerations, we will revise the manuscript to further elaborate on these distinctions and better motivate our contributions by contextualizing them within the limitations of prior LLM-based optimization methods.

---

> ### Author Response · Authors · 2024-11-20
> **Response to Reviewer BtD7 (2)**
>
> > * My second concern is regarding the choice of baselines. While I appreciate the inclusion of several standard SR methods, I think the comparisons do not precisely show the benefits of the paper's contributions.
> (a) Given prior works such as [1, 2, 3] in LLM-based SR/optimization, it would have provided utility to see the performance of the proposed method in comparison to its closest counterparts. Alternatively, if the argument is that some of the ablations subsume these methods, it would then be useful to see those ablations presented as the main baselines on all the datasets instead of only on a partial set.
> >
>
> Thanks for the thoughtful suggestion.
> In response to your comment for baseline comparisons, we have conducted additional experiments with LMX [1] and FunSearch [2] using their original implementations and hyperparameters. For LMX, we directly utilized their open-source code, and for FunSearch, we adapted the prompt and feedback design to suit the equation-as-program task. We did not include OPRO [3] in these experiments as its original framework is designed for very different optimization tasks (such as prompt optimization) and would require substantial modifications to be used for SR. For a fair comparison, we run these LLM-based baselines for the same number of iterations as LLM-SR. Due to the limited rebuttal time, we were only able to conduct experiments on the oscillation 1 and oscillation 2 datasets with the GPT-3.5 LLM backbone.
>
>
> Results (please see the table below) show that LLM-SR outperforms both LMX and FunSearch in in-domain and OOD test settings. Notably, when skeleton+optimizer design is ablated (allowing the LLM to generate complete equations without placeholder parameter vectors followed by optimization), performance drops significantly but still outperforms LMX and FunSearch. Also, we observe that these LLM-based baselines obtain worse performance than the SR baselines (DSR, uDSR, PySR) on both datasets. This underscores our focus on comparison with SR baselines to showcase our framework's strengths against state-of-the-art models of the task. We also observe that the incorporation of both 'prior knowledge' and the 'programming' components enables LLM-SR to consistently outperform leading SR baselines in both in-domain and OOD test scenarios.
>
>
> We agree that including these LLM-based comparisons strengthen the paper's positioning. We will experiment on the remaining datasets and will incorporate the corresponding results and a detailed discussion of LLM-based baselines in the updated manuscript.
>
>
>
>
> | Model        | Oscillation 1 (ID/OOD) | Oscillation 2 (ID/OOD) |
> |----------------------------|----------------------|----------------------|
> | DSR        | 0.0087 / 0.2454       | 0.0580 / 0.1945       |
> | uDSR        | 0.0003 / 0.0007       | 0.0032 / 0.0015       |
> | PySR        | 0.0009 / 0.3106       | 0.0002 / 0.0098       |
> | LMX [1]         | 0.5031 / 48.93       | 1.004 / 0.9371       |
> | FunSearch [2]         | 0.4840 / 8.059       | 0.7234 / 0.5861       |
> | LLM-SR (w/o skeleton+optimizer)  | 0.1371 / 0.6764       | 0.3780 / 0.3751       |
> | LLM-SR (w/o Prior)  | 0.0001 / 0.0029       | 4.65e-5 / 0.0071       |
> | LLM-SR (w/o Program)  | 0.0001 / 0.0035       | 3.22e-5 / 0.0043       |
> | LLM-SR  | 7.89e-8 / 0.0002       | 2.12e-7 / 3.81e-5       |
>
> ---
> > * (b) The ablation results from Figure 6 show that a substantial drop in performance occurs when coefficient optimization is removed. Based on this, it would have been useful to see whether simply equipping the non-LLM baselines with a similar coefficient optimizer would close the gaps that we currently see in Table 1. I am currently not convinced that the LLM is needed to facilitate the evolutionary search of equation templates.
> >
>
> Thanks for the insightful question. We would like to clarify that **non-LLM symbolic regression baselines (expect E2E) treat SR as a structure followed by coefficient optimization task instead of directly optimizing both structure and coefficients simultaneously**. Therefore, the suggestion that _"equipping the non-LLM baselines with a similar coefficient optimizer would close the gaps"_ is not applicable as non-LLM SR baselines are already equipped with the coefficient optimization process and this is not something that is unique to our LLM-based framework.
>
> We understand that the term "w/o coeff optimization" in Figure 6 might have been confusing. To avoid misinterpretation, we will revise it to "w/o skeleton+optimizer," which better reflects the distinction. We hope our response addresses your concern. Please let us know if we have correctly interpreted your concern or if further clarification is needed.

---

> ### Author Response · Authors · 2024-11-20
> **Response to Reviewer BtD7 (3)**
>
> **Response to Questions:**
>
> > 1. L243 briefly described how population initialization is performed. Is the linear equation skeleton manually written? How many skeletons are provided at initialization?
> > 2. In L149, should the expectation be over the data points in D?
> > 3. In Appendix E, could you clarify what LLM-SR (w/o Prior) refers to? From the text described in the main paper, I could not find mention of the described "problem-specific prior knowledge" (L1220).
> >
> Thank you for your thoughtful questions. Please find our answers below.
>
> 1. Yes, the linear skeleton is manually provided as an initial example in the initial prompt (as shown in Figure 2).  The motivation of having the simplest linear design for the initial example is to avoid manual knowledge insertion and let LLM to mainly use its domain knowledge about the equation structures. Only one such linear skeleton is provided at the initial prompt. All islands in the experience buffer (populations) are initialized with this example and are subsequently updated through the knowledge-guided, LLM-generated hypotheses during iterative search.
>
> 2. You are correct. We will update the notation in the manuscript.
>
> 3. The term "LLM-SR (w/o Prior)" in Figure 17 and Appendix E corresponds to "w/o problem specification" in Figure 6. This variant evaluates LLM-SR’s performance when natural language descriptions of the problem and the physical variables are removed from the prompt. This ablation highlights the importance of domain-specific prior knowledge for LLM-SR’s performance. We will update the terminology to be consistent throughout the manuscript.
>
>
> ---
> **Response to Minor Comments:**
>
> > * The choice of describing the contents in Section 2.4 as "Experience Management" is mildly concerning. Given a large body of prior work in evolutionary computation, it would be appropriate to avoid fragmenting the literature for the benefit of differentiation and instead re-use existing terminology. If there is indeed an aspect that warrants a new term, please do correct me.
> >
> Thank you for the suggestion. The term "Experience Management" was inspired by concepts like "memory/experience mechanism" commonly used in LLM agent frameworks (e.g. Zhang et al. 2024). These terms usually reflect the mechanism of storing diverse successful hypotheses from previous iterations, which guide the LLM iterative refinement. We are open to revising the terminology to better align with existing evolutionary search literature. If you have specific suggestions for a more appropriate term, we would greatly appreciate your input.
>
> Zhang et al., A Survey on the Memory Mechanism of Large Language Model-based Agents, 2024
>
>
> > * It was mildly odd to read that mathematical equations have been "unreasonably" effective in describing complex phenomena. Unclear why we think it is unreasonable.
> >
> Thank you for the feedback. The phrasing was inspired by Eugene Wigner's remark on the "unreasonable effectiveness of mathematics in the natural sciences." (Wigner , E.P. (1960))

---

> ### Author Response · Authors · 2024-11-22
> **Looking forward to discussion**
>
> Dear Reviewer BtD7,
>
> We hope our answers and new experiments have addressed your concerns and questions. Please let us know if you have any more questions before the end of the discussion period.
>
> Should there be no additional concerns, we kindly ask you to consider revising your score.
>
> Thank you for your time and thoughtful feedback!

---

> > ### Comment · Reviewer_BtD7 · 2024-11-25
> >
> > > In light of these considerations, we will revise the manuscript to further elaborate on these distinctions and better motivate our contributions by contextualizing them within the limitations of prior LLM-based optimization methods.
> >
> > I am taking on good faith that the authors will indeed make this change as no revision PDF has been uploaded. I believe it is very important to transparently provide the reader with the specific new contribution of this work in light of previous work from the outset.
> >
> > It is good to see the direct comparisons with the new baselines. Thank you for running the experiments.
> >
> > > Therefore, the suggestion that "equipping the non-LLM baselines with a similar coefficient optimizer would close the gaps" is not applicable as non-LLM SR baselines are already equipped with the coefficient optimization process and this is not something that is unique to our LLM-based framework.
> >
> > I see. In that case, I would urge the authors to make this point clear in their revised manuscript.
> >
> > > We are open to revising the terminology to better align with existing evolutionary search literature. If you have specific suggestions for a more appropriate term, we would greatly appreciate your input.
> >
> > I would like to see a connection drawn to the notion of a population in evolutionary computation. If the authors prefer not changing the terminology, please include a line in the text drawing this connection.
> >
> > > The phrasing was inspired by Eugene Wigner's remark on the "unreasonable effectiveness of mathematics in the natural sciences." (Wigner , E.P. (1960))
> >
> > Please add the citation, in that case.
> >
> > ---
> >
> > Overall, thank you for the comprehensive responses to my feedback! Most of my concerns have been addressed. I am increasing my score while reducing the confidence by a notch to account for some uncertainty in my assessment.

---

> > > ### Author Response · Authors · 2024-11-25
> > > **Thank you from Authors**
> > >
> > > Thank you once again for your time and valuable insights. We are glad that our rebuttal has resolved your concerns and appreciate the raised score. We will make sure to update the manuscript in line with your suggestions and incorporate changes in the camera-ready version.

---

### Official Review · Reviewer_89Tg · 2024-11-04

**Soundness:** 4
**Presentation:** 3
**Contribution:** 3
**Rating:** 8
**Confidence:** 4

**Summary:**

The paper introduces **LLM-SR**, a novel framework for discovering scientific equations using **Large Language Models (LLMs)**. The approach integrates LLMs' scientific knowledge and code generation capabilities with evolutionary search and optimization techniques to generate, evaluate, and refine mathematical equation hypotheses iteratively. The key steps involve generating equation skeletons, optimizing parameters using Python-based tools, and managing an experience buffer to iteratively improve the search process.

**Main Contributions:**
1. **Novel Framework**: LLM-SR combines LLMs' strengths with evolutionary algorithms to navigate complex equation discovery.
2. **Integration of Scientific Priors**: The method leverages LLMs' embedded scientific knowledge for more efficient hypothesis generation.
3. **Benchmark Creation**: The authors design new benchmark problems in physics, biology, and materials science to test the method's capabilities and avoid memorization of well-known equations.
4. **Performance Advantage**: LLM-SR reportedly outperforms traditional symbolic regression methods, showing better accuracy and generalization, especially in out-of-domain tests.
5. **Ablation Study**: Demonstrates the importance of various components, such as problem specification and iterative refinement, in the overall performance of LLM-SR.

**Strengths:**

1. **Innovative Application of LLMs for Scientific Equation Discovery**:
   - The authors introduce an innovative approach by leveraging the scientific knowledge and code generation capabilities of large language models (LLMs) for equation discovery. Representing equations as code and integrating data-driven feedback and optimization provides a novel perspective for scientific modeling.

2. **Experience Buffer and Iterative Optimization**:
   - The use of an experience buffer to maintain high-quality equation examples and facilitate iterative optimization is a unique feature. This mechanism helps the model learn and recall effective generation patterns, improving the quality and efficiency of equation exploration.

3. **Comprehensive Multi-Domain Benchmarking**:
   - The paper demonstrates the method’s applicability across multiple scientific domains, including physics, biology, and materials science. The newly designed benchmarks, which aim to prevent memorization of known equations, reflect the authors' commitment to scientific rigor.

4. **Promising Experimental Results**:
   - Despite certain challenges and areas for improvement, LLM-SR shows promising performance compared to existing symbolic regression methods, particularly in out-of-domain tests. This indicates potential for tackling complex scientific problems and suggests a valuable direction for future research.

**Weaknesses:**

**Complexity and Necessity of the Method**:
The authors introduce complex mechanisms such as sampling strategies and an island model, showcasing innovative efforts. However, clearer justification is needed to explain the necessity of these elements and their specific contributions to enhancing the method’s performance. Simplifying and elucidating these mechanisms' actual benefits would improve the clarity and persuasiveness of the method.

**Utilization and Verification of Prior Knowledge**:
Although the authors assume that the LLM leverages its scientific prior knowledge to generate reasonable equations, the paper lacks detailed analysis and experimental validation of how this mechanism functions and its tangible effects. Further exploration and evidence would more convincingly support this claim and make the method’s core advantage clearer to readers.

**Breadth and Realism of Experimental Design**:
The current experiments are based on data generated from known equations, providing a basic evaluation. However, data in real-world scientific research often contains noise and outliers, and the method’s performance under such conditions is a critical indicator of its practical applicability. Including datasets with measurement errors and outliers would help demonstrate the method’s robustness and generalizability in complex, realistic scenarios.

**Impact of Programming Capabilities on Experimental Results**:
The paper emphasizes the LLM's ability to generate programming code but lacks analysis of whether these programming capabilities interfere with the experimental results. The boundary between the LLM’s programming skills and scientific reasoning remains unclear, which might mean that experimental outcomes reflect coding proficiency rather than pure scientific inference. Clarifying the role of these aspects in equation generation would aid in better understanding the method’s true strengths and limitations.

**Questions:**

1. **Clarification on the Necessity of Complex Mechanisms**:
   - *Question*: Could the authors elaborate on the rationale behind incorporating complex mechanisms such as the sampling strategy and island model? How do these elements contribute to the overall performance, especially compared to simpler alternatives?
   - *Suggestion*: Including comparative results or ablation studies that illustrate the impact of these mechanisms on the method’s performance would be helpful to better understand their necessity and specific contributions.

2. **Verification of LLM Prior Knowledge Utilization**:
   - *Question*: How do the authors verify that the LLM’s embedded scientific prior knowledge is effectively utilized during the equation generation process? Are there examples or analyses demonstrating cases where this prior knowledge played a significant role?
   - *Suggestion*: We suggest adding controlled experiments that isolate and assess the contribution of prior knowledge. Comparing models with and without domain-specific pretraining might further strengthen this aspect.

3. **Evaluation on Real-World Datasets with Noise and Outliers**:
   - *Question*: Do the authors plan to evaluate LLM-SR on more challenging, real-world datasets that contain noise, outliers, or measurement errors? How do they foresee the method performing under such conditions?
   - *Suggestion*: Expanding the experiments to include data with inherent noise and outliers, or discussing the method’s anticipated robustness in these cases, could help demonstrate its applicability to complex, real-world scenarios.

4. **Impact of LLM Programming Performance on Results**:
   - *Question*: Have the authors considered examining the extent to which the LLM’s programming capabilities, rather than scientific reasoning, might influence the results? How do they ensure that the outcomes reflect genuine equation discovery rather than sophisticated code generation?
   - *Suggestion*: Providing an analysis or qualitative examples that differentiate between the model’s programming skills and its scientific reasoning in generating equations would be valuable to clarify the contributions and limitations of the approach.

5. **Interpretability of Results**:
   - *Question*: The authors state that the generated equations are more interpretable and scientifically meaningful. Could they provide examples or a detailed comparison showing how LLM-SR’s output stands out in terms of interpretability relative to baseline methods?
   - *Suggestion*: Presenting case studies or examples where the equations discovered by LLM-SR provide clearer or more scientifically insightful interpretations would reinforce this claim.

6. **Broader Comparison with Hybrid Methods**:
   - *Question*: How does LLM-SR compare with recent hybrid approaches that incorporate domain knowledge or leverage deep learning for symbolic regression? Are there particular strengths or limitations of LLM-SR compared to these methods?
   - *Suggestion*: Including comparisons with other state-of-the-art hybrid methods or discussing potential improvements and challenges in adapting LLM-SR could provide more context on its positioning and competitive advantages.

**Details Of Ethics Concerns:**

For **ethical considerations**, we recommend the authors provide the following details:

- **Licenses and Usage Permission Details for Datasets**:
   - *Suggestion*: We encourage the authors to report the type of licenses and usage permissions associated with the datasets used in the study. Clarifying whether each dataset is publicly available and whether its use complies with the original publisher’s license agreement would enhance the transparency and compliance of the paper.
   - *Rationale*: This is particularly important in academic research to ensure that the use of datasets does not violate copyright, privacy, or other relevant legal considerations. Clear documentation of dataset usage can help prevent potential ethical issues and provide future researchers with well-defined guidelines.

---

> ### Author Response · Authors · 2024-11-20
> **Response to Reviewer 89Tg (1)**
>
> Thank you for your insightful and constructive feedback on our paper. Please find our responses below.
>
> ---
> > * Question: Could the authors elaborate on the rationale behind incorporating complex mechanisms such as the sampling strategy and island model? How do these elements contribute to the overall performance, especially compared to simpler alternatives?
> > * Suggestion: Including comparative results or ablation studies that illustrate the impact of these mechanisms on the method’s performance would be helpful to better understand their necessity and specific contributions.
> >
>
> Thank you for your question. As we also mentioned in the paper (please see sec. 2.4, Ln 223-226), the multi-island design and corresponding sampling strategy are motivated by successful prior works in symbolic regression, particularly PySR (Cranmer et al. 2023), which demonstrated the benefits of multi-island evolution in enabling diverse exploration and robustness for equation search.
>
> In response to the reviewer's suggestion, we have conducted additional ablation experiments to analyze the impact of multi-island and corresponding sampling design on LLM-SR. Specifically, we removed the multi-island buffer design (i.e., we keep only one island in the buffer) and used a simpler deterministic top-k selection approach for in-context example selection. We ran this model variant for the same number of iterations (~2500) as other LLM-SR runs, and benchmarked it on the oscillation 2 and stress-strain datasets with the GPT-3.5 LLM backbone.
>
> The results (shown in the table below) demonstrate that the multi-island design positively impacts LLM-SR performance in both in-domain and OOD test settings (reported as NMSE, consistent with Table 1). Qualitative analysis reveals that the number of islands plays a critical role in balancing exploitation and exploration within LLM-SR. Specifically, with fewer islands, the framework exhibits reduced exploration, generating less diverse equation hypotheses and converging prematurely to equation structures produced in the early iterations.
>
> | Model Variant        | Stress-Strain (ID/OOD) | Oscillation 2 (ID/OOD) |
> |----------------------------|----------------------|----------------------|
> | LLM-SR (w/o multi-island & sampling)          | 0.0257 / 0.1010       | 6.23e-6 / 0.0008       |
> | LLM-SR  | 0.0210 / 0.0516       | 2.12e-7 / 3.81e-5       |
>
> We will incorporate these results and discussions in the final revision of manuscript.
>
> ---
> > * Question: How do the authors verify that the LLM’s embedded scientific prior knowledge is effectively utilized during the equation generation process? Are there examples or analyses demonstrating cases where this prior knowledge played a significant role?
> > * Suggestion: We suggest adding controlled experiments that isolate and assess the contribution of prior knowledge.
> >
>
> Thank you for the question. We have already studied this in our current ablation experiments. Specifically, the "w/o problem specification" ablation in Figure 6 (and "w/o Prior" in Figure 17) demonstrate LLM-SR's performance when problem and variable descriptions are removed from the prompt. By removing these contexts from the prompt, we control for the model’s use of prior knowledge in the hypothesis generation.
> In this *No Prior* setting, the model variant consistently performs worse across all datasets (see Figure 17), underscoring the critical role of domain knowledge in equation discovery.
>
> Beyond the ablation results, qualitative analysis of the discovered equations (Sec 3.4 and Figure 4) highlights how domain-specific prior knowledge in LLM-SR facilitated identifying correct forms, including various physical terms, in problems like nonlinear oscillators. This observation extends to other benchmark problems as well, as detailed in Appendix G (pg. 24–29).
>
> We will make sure to clarify these points and add more discussion about them in the revised version of manuscript.

---

> ### Author Response · Authors · 2024-11-20
> **Response to Reviewer 89Tg (2)**
>
> > * Question: Do the authors plan to evaluate LLM-SR on more challenging, real-world datasets that contain noise, outliers, or measurement errors? How do they foresee the method performing under such conditions?
> > * Suggestion: Expanding the experiments to include data with inherent noise and outliers, or discussing the method’s anticipated robustness in these cases, could help demonstrate its applicability to complex, real-world scenarios.
> >
>
> Thanks for the suggestion. We agree that testing on noisy, challenging data is crucial for evaluating these methods' applicability in actual scientific discovery. Regarding the reviewer's question, we want to emphasize two key points:
>
> First, our current "stress-strain dataset" inherently contains measurement noise and outliers from **real experimental observations** (detailed in Appendix D.3). The results in Table 1 demonstrate LLM-SR's better performance on this experimental dataset compared to the baselines.
>
> Second, following the reviewer's suggestion, we conducted **additional experiments by introducing controlled Gaussian noise** at varying levels to the oscillation 2 benchmark problem, comparing LLM-SR against PySR (our leading baseline). Results of these new experiments, visualized in Appendix F (Figure 18), reveal that while noise universally challenges equation discovery, LLM-SR demonstrates significantly better robustness compared to PySR. This is particularly evident even at moderate noise levels (σ = 0.01), where LLM-SR maintains excellent performance (NMSE: 2.00e-4 ID, 7.00e-4 OOD) while PySR's performance degrades substantially (NMSE: 6.20e-1 ID, 7.67e-1 OOD). This aligns with our expectations: as data quality decreases, the importance of effective priors grows, amplifying the benefits of incorporating prior domain knowledge through LLM-SR.
>
> ---
> > * Question: Have the authors considered examining the extent to which the LLM’s programming capabilities, rather than scientific reasoning, might influence the results? How do they ensure that the outcomes reflect genuine equation discovery rather than sophisticated code generation?
> > * Suggestion: Providing an analysis or qualitative examples that differentiate between the model’s programming skills and its scientific reasoning in generating equations would be valuable to clarify the contributions and limitations of the approach.
> >
>
> We appreciate the reviewer's question and would like to clarify that this ablation has already been studied in our current experiments. Specifically, the *"w/o Equation as Program"* ablation in Figure 6 (and *"w/o Program"* in Figure 17) show LLM-SR's performance when equation hypothesis generation is restricted to mathematical expressions rather than Python programming functions. To be more specific, the ablation presented in Figure 17 helps us disentangle the role of programming and scientific domain knowledge by showing that both components contribute significantly to the model's success, yet neither component alone can achieve optimal performance. We will make sure to clarify these points and add more discussion about them in the revised version of the manuscript.
>
> ---
> > * Question: The authors state that the generated equations are more interpretable and scientifically meaningful. Could they provide examples or a detailed comparison showing how LLM-SR’s output stands out in terms of interpretability relative to baseline methods?
> > * Suggestion: Presenting case studies or examples where the equations discovered by LLM-SR provide clearer or more scientifically insightful interpretations would reinforce this claim.
> >
>
> Thank you for the comment. We would like to clarify that Figure 4 and Figures 20-24 in Appendix G already provide qualitative examples of the discovered equations and demonstrate the interpretability gains from LLM-SR vs the baselines. For example, in both oscillation problems of Figure 4, LLM-SR identifies the equation structure as a combination of driving force, damping force, and restoring force terms, relating them to the problem's physical characteristics. In contrast, baselines (DSR, uDSR, PySR) generate equations lacking interpretability and understanding of the physical meanings or relations behind the terms. The equations appear as a combination of mathematical operations and variables without clear connection to the problem's underlying principles.

---

> ### Author Response · Authors · 2024-11-20
> **Response to Reviewer 89Tg (3)**
>
> > * Question: How does LLM-SR compare with recent hybrid approaches that incorporate domain knowledge or leverage deep learning for symbolic regression? Are there particular strengths or limitations of LLM-SR compared to these methods?
> > * Suggestion: Including comparisons with other state-of-the-art hybrid methods or discussing potential improvements and challenges in adapting LLM-SR could provide more context on its positioning and competitive advantages.
> >
>
> Thank you for the suggestion. The baselines evaluated in this study are state-of-the-art for symbolic regression. Among these baselines, E2E (Kamienny et al. 2022) and NeSymReS (Biggio et al. 2021) are deep learning pre-trained transformer-based models for symbolic regression; DSR (Petersen et al. 2021) is a reinforcement learning model with neural policy design for equation generation; and uDSR (Landajuela et al. 2022) is the hybrid advanced version of DSR with genetic programming combined with reinforcement learning and transformer neural policy design. PySR (Cranmer et al. 2023) is also one of the recent most popular methods for this task which is based on the multi-island evolutionary search design.
>
> We would like to clarify that none of these methods use prior scientific knowledge in their equation generation/search process, which is one of the main motivations of LLM-SR. Please let us know if we have correctly interpreted your concern or if you have any specific suggestion for the new baselines to be included in the experiments.
>
> ---
> > * For ethical considerations, we recommend the authors provide the following details: Licenses and Usage Permission Details for Datasets:
> > * Suggestion: We encourage the authors to report the type of licenses and usage permissions associated with the datasets used in the study. Clarifying whether each dataset is publicly available and whether its use complies with the original publisher’s license agreement would enhance the transparency and compliance of the paper.
> > * Rationale: This is particularly important in academic research to ensure that the use of datasets does not violate copyright, privacy, or other relevant legal considerations. Clear documentation of dataset usage can help prevent potential ethical issues and provide future researchers with well-defined guidelines.
> >
>
> Thank you for the suggestion. In this study, we explored LLM-SR across four benchmark problems, three of which were developed and introduced in this work itself. These benchmarks were generated by sampling data from underlying functions or through numerical simulations. We plan to release these datasets publicly under a Creative Commons license. The fourth benchmark, the material behavior problem, was introduced in prior work, and its data is publicly available at https://data.mendeley.com/datasets/rd6jm9tyb6/1 under the CC BY 4.0 license. These datasets are already provided in the anonymous repository link shared in our manuscript.

---

> ### Author Response · Authors · 2024-11-22
> **Looking forward to discussion**
>
> Dear Reviewer 89Tg,
>
> We hope our answers and new experiments have addressed your concerns and questions. Please let us know if you have any more questions before the end of the discussion period.
>
> Should there be no additional concerns, we kindly ask you to consider revising your score.
>
> Thank you for your time and thoughtful feedback!

---

> ### Author Response · Authors · 2024-11-26
> **Reminder from Authors**
>
> Dear Reviewer 89Tg,
>
> Thank you for your feedback during the review process!
> We believe that our detailed response has addressed your concerns. If you have any concerns or questions, please do not hesitate to let us know before the author discussion period ends. We will be happy to answer them during the discussion.
>
> Thank you,

---

> > ### Comment · Reviewer_89Tg · 2024-11-27
> >
> > My concerns have been addressed, and the work is both thorough and insightful. I suggest that the committee consider this work as a candidate for the best paper award.

---

> > > ### Author Response · Authors · 2024-11-28
> > > **Thank you from Authors**
> > >
> > > Thank you once again for your time and valuable feedback! We are glad that our rebuttal has addressed your concerns and deeply appreciate the raised score and the best paper nomination!
> > >
> > > Regarding the ethics review flag, please let us know if you require any further information. If there are no additional concerns, we appreciate the change.

---

### Official Review · Reviewer_4Qbb · 2024-11-08

**Soundness:** 4
**Presentation:** 4
**Contribution:** 4
**Rating:** 8
**Confidence:** 3

**Summary:**

This paper introduces LLM-SR, a new approach to discovering mathematical equations from data using LLMs' scientific knowledge and code generation. LLM-SR treats equations as program structures. LLM-SR achieves significant improvements over baseline symbolic regression methods.

**Strengths:**

1. Innovative approach: It is innovative to use LLMs for scientific equation discovery and symbolic regression. LLM-SR integrates LLMs' prior scientific knowledge and and code generation ability to achieve good performance.

2. Robust evaluation: The authors have carefully designed benchmark problems that align well with the real-world applications.

3. Comprehensive analysis: The qualitative analysis effectively evaluates the model's OOD generalizability  and the ablation study provides insight into the contribution of each of model's component.

4. Well written: The paper is well written with clear data presentation.

**Weaknesses:**

The paper does not provide detailed insight into how LLM-SR handles edge cases, such as noisy data, or highly complex functions. This could be important for real-world applications where data may be imperfect or equations are inherently intricate.

**Questions:**

1. A more thorough analysis of the computational cost, especially comparing LLM-SR to traditional symbolic regression techniques, would help gauge the practical feasibility of this approach.

2. Providing a real-world example of how LLM-SR assists in scientific problem solving could make the results more compelling.

---

> ### Author Response · Authors · 2024-11-20
> **Response to Reviewer 4Qbb**
>
> Thank you for your thoughtful comments. Please find our responses below.
>
> > * The paper does not provide detailed insight into how LLM-SR handles edge cases, such as noisy data, or highly complex functions. This could be important for real-world applications where data may be imperfect or equations are inherently intricate.
> > * Providing a real-world example of how LLM-SR assists in scientific problem solving could make the results more compelling.
> >
>
> Thank you for the insightful feedback. The data quality and complexity of the underlying functions indeed present key challenges for symbolic regression. We would like to note that we have tried to design and select problems that are relatively complex and simulate the real-world discovery problems. This includes a **real experimental dataset** (stress-strain measurements) that evaluates the applicability of SR methods on real measurements. We observe that LLM-SR outperforms state-of-the-art SR baselines on such datasets, showing its potential in handling complex, real-world problems.
>
> Also, we have conducted **new experiments by adding Gaussian noise** at different levels to the data and evaluating LLM-SR as well as PySR (as the leading baseline) on the oscillation 2 benchmark problem. We have provided these results, highlighted in blue, in Appendix F (Figure 18). The results indicate that while increasing levels of noise introduce challenges in equation discovery, LLM-SR is affected substantially less than PySR. We observe that with moderate amounts of noise (σ = 0.01), LLM-SR maintains its performance (NMSE: 2.00e-4 in-domain, 7.00e-4 OOD) while PySR's performance degrades significantly (NMSE: 6.20e-1 in-domain, 7.67e-1 OOD), and this robustness gap persists at higher noise levels (σ = 0.1). These observations confirm that with lower quality of data, the role of effective priors becomes more important, leading to more significant benefits from incorporating prior knowledge in LLM-SR.
>
>
> > * A more thorough analysis of the computational cost, especially comparing LLM-SR to traditional symbolic regression techniques, would help gauge the practical feasibility of this approach.
> >
>
> Thank you for the suggestion. Given the foundational differences between SR methods in their approaches, a direct comparison of computational costs is challenging. We have provided detailed implementation specifications for both LLM-SR and baseline methods in Appendices A and B. While traditional search-based SR baselines require over 2M iterations to converge to their best performance, LLM-SR achieves superior results within approximately 2.5K iterations.
> The computational requirements of LLM-SR vary with the choice of language model backbone. In our experiments with the Mixtral-8x7B model, we utilize 4 NVIDIA RTX 8000 GPUs (48GB memory each). For experiments with GPT-3.5-turbo, we leverage OpenAI API. Under these configurations, we observe that LLM-SR achieves faster wall-clock time to solution compared to prominent search-based SR methods like PySR and uDSR. We will add a remark on this in the revised paper.

---

> ### Author Response · Authors · 2024-11-22
> **Looking forward to discussion**
>
> Dear Reviewer 4Qbb,
>
> We hope our answers and new experiments have addressed your concerns and questions. Please let us know if you have any more questions before the end of the discussion period.
>
> Thank you for your time and thoughtful feedback!

---

### Author Response · Authors · 2024-11-20
**General Response to Reviewers**

We sincerely thank all the reviewers for their time and expertise in evaluating our manuscript. We are grateful for the positive feedback, including the recognition of LLM-SR as a novel and innovative contribution (Revs. 4Qbb, 89Tg, BtD7), the acknowledgment that our benchmark problems are well-motivated (Revs. 4Qbb, BtD7), and the appreciation for the robustness, rigor, and comprehensiveness of our experiments and analyses (Revs. 4Qbb, 89Tg).
The constructive feedback provided by the reviewers has been invaluable in improving the paper. We have provided detailed responses to each reviewer's comments individually and outline below the key results from new experiments conducted in response to multiple reviewer requests.

**New Experiments**:

- **Comparison with other LLM baselines for SR:** As per reviewer BtD7’s suggestion, we have conducted experiments to compare LLM-SR with LLM-based optimization baselines. The results indicate that LLM-SR, by incorporating prior knowledge, programming capabilities, data-driven parameter optimization, and dynamic memory design outperforms such baselines.

- **Multi-island ablation:** As per reviewer 89Tg’s suggestion, we performed an ablation study on the effect of multi-island design and sampling strategies. The results show that multi-island design helps LLM-SR by maintaining a diverse population of high-quality hypotheses.

- **GPT-4o Backbone:** As per reviewer AFCG's suggestion, we evaluated LLM-SR using GPT-4o as the backbone model to assess the impact of larger LLMs. Our findings indicate that LLM-SR exhibits improved performance when paired with GPT-4o.

- **Robustness to Noise:** We introduced different levels of noise to the data and benchmarked LLM-SR against the leading baseline. The results demonstrate that LLM-SR is significantly more robust to noise compared to its counterparts due to the incorporation of prior knowledge.

We are happy to answer any further questions/concerns.

---

### Comment · Area_Chair_wvwy · 2024-11-25

Dear Reviewers,

This is a kind reminder that the dicussion phase will be ending soon on November 26th. Please read the author responses and engage in a constructive discussion with the authors.

Thank you for your time and cooperation.

Best,

Area Chair

---

### Meta-Review · Area_Chair_wvwy · 2024-12-19

**Metareview:**

This paper proposes LLM-SR, a framework that leverages large language models for symbolic regression by integrating program synthesis, numerical optimization, and evolutionary search to discover accurate and generalizable scientific equations. Most of the reviewers agreed that the proposed approach is novel and technically solid. Moreover, this paper is written with clear data presentation and effective use of visualizations.

During the review process, Reviewer 89Tg raised ethical concerns regarding dataset usage. The authors addressed this in their rebuttal by explaining the licensing and availability of the datasets, with detailed information provided in the paper's appendix. The reviewer confirmed that their concerns had been addressed and suggested the paper for the best paper award. As a result, I believe further ethical review is unnecessary. Nevertheless, I strongly recommend that the authors further discuss dataset usage in the final version to enhance clarity and transparency.

All the reviews tend to accept the paper during the reviewer-author discussion phase. Therefore, I recommend this paper to the ICLR 2025 conference.

**Additional Comments On Reviewer Discussion:**

Reviewers 4Qbb, 89Tg, BtD7, and AFGC rated this paper as 8: accept.

The reviewers raised the following concerns.

- Scalability (rasied by Reviewer 4Qbb, 89Tg and AFGC).

- Insufficient experiments (raised by Reviewer 89Tg and BtD7).

- Limited novelty (raised by Reviewer BtD7).

- Lack of real-world cases (raised by Reviewer 4Qbb).

- Ethical concerns regarding dataset usage (raised by Reviewer 89Tg).

During the review process, Reviewer 89Tg raised ethical concerns regarding dataset usage. The authors addressed this in their rebuttal by explaining the licensing and availability of the datasets, with detailed information provided in the paper's appendix. The reviewer confirmed that their concerns had been addressed and suggested the paper for the best paper award.As a result, I believe further ethical review is unnecessary. Nevertheless, I strongly recommend that the authors further discuss dataset usage in the final version to enhance clarity and transparency.


Furthermore, by conducting additional experiments and providing more details in the rebuttal, the authors address all concerns raised by the responding reviewers 89Tg, BtD7, and AFGC.

Therefore, I will recommend accepting this paper in its current state.

---

### Decision · Program_Chairs · 2025-01-22

Accept (Oral)